# Curvilinear Distance Metric Learning

**Shuo Chen[†], Lei Luo[‡*], Jian Yang[†*], Chen Gong[†], Jun Li[§], Heng Huang[‡]**

## Abstract

Distance Metric Learning aims to learn an appropriate metric that faithfully measures the distance between two data points. Traditional metric learning methods usually calculate the pairwise distance with fixed distance functions (*e.g.,* Euclidean distance) in the projected feature spaces. However, they fail to learn the underlying geometries of the sample space, and thus cannot exactly predict the intrinsic distances between data points. To address this issue, we first reveal that the traditional linear distance metric is equivalent to the cumulative arc length between the data pair's nearest points on the learned straight measurer lines. After that, by extending such straight lines to general curved forms, we propose a Curvilinear Distance Metric Learning (CDML) method, which adaptively learns the nonlinear geometries of the training data. By virtue of Weierstrass theorem, the proposed CDML is equivalently parameterized with a 3-order tensor, and the optimization algorithm is designed to learn the tensor parameter. Theoretical analysis is derived to guarantee the effectiveness and soundness of CDML. Extensive experiments on the synthetic and real-world datasets validate the superiority of our method over the state-of-the-art metric learning models.

## 1 Introduction

The goal of a Distance Metric Learning (DML) algorithm is to learn the distance function for data pairs to measure their similarities. The learned distance metric successfully reflects the relationships within data points and significantly improves the performance of many subsequent learning tasks, such as classification [22], clustering [23], retrieval [24], and verification [14]. It has recently become an active research topic in machine learning community [30, 29].

The well-studied DML methods are usually linear, namely Mahalanobis distance metric based models [23]. Under the supervisions of pairwise similarities, they intend to learn a Semi-Positive-Definite (SPD) matrix $M = PP^\top \in \mathbb{R}^{d \times d}$ to decide the squared parametric distance $\text{Dist}_P^2(x, \widehat{x}) = (x - \widehat{x})^\top M (x - \widehat{x})$ between data points $x$ and $\widehat{x}$ in the $d$-dimensional space. It is notable that such a linear Mahalanobis distance is equivalent to the Euclidean distance in the $m$-dimensional feature space projected by $P \in \mathbb{R}^{d \times m}$. To perform the learning of the parameter $M$, intensive efforts have been put to design various loss functions and constraints in optimization models. The early works Large Margin Nearest Neighbor (LMNN) [27] and Information-Theoretic Metric Learning (ITML) [7] utilized the must-link and cannot-link information to constrain the ranges of the learned distances. Instead of fixing the distance ranges in the objective, Geometric Mean Metric Learning (GMML) [31]

---

[†]S. Chen, J. Yang, and C. Gong are with the PCA Lab, Key Lab of Intelligent Perception and Systems for High-Dimensional Information of Ministry of Education, and Jiangsu Key Lab of Image and Video Understanding for Social Security, School of Computer Science and Engineering, Nanjing University of Science and Technology, Nanjing 210094, China (E-mail: {shuochen, csjyang, chen.gong}@njust.edu.cn).

[‡]L. Luo and H. Huang are with the Electrical and Computer Engineering, University of Pittsburgh, and also with JD Finance America Corporation, USA (E-mail: lel94@pitt.edu, henghuanghh@gmail.com).

[§]J. Li is with the Institute for Medical Engineering & Science, Massachusetts Institute of Technology, Cambridge, MA, USA (E-mail: junli@mit.edu).

[*]L. Luo and J. Yang are corresponding authors.

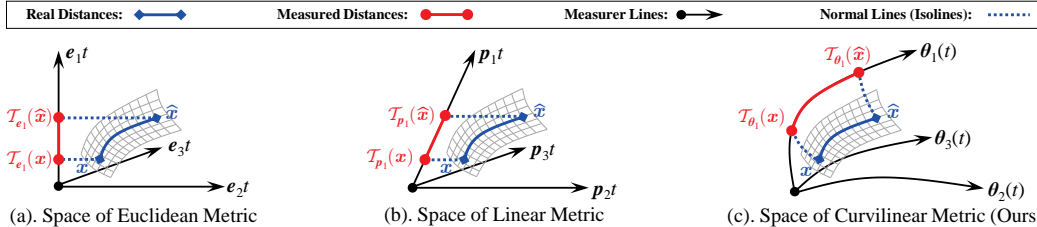

(a). Space of Euclidean Metric      (b). Space of Linear Metric      (c). Space of Curvilinear Metric (Ours)

Figure 1: Conceptual illustrations of Euclidean metric, linear (Mahalanobis) metric, and our proposed curvilinear metric in three-dimensional space. For a pair of data points (*i.e.,* $x$ and $\widehat{x}$) from the spatial surface, metrics find out the nearest (calibration) points (*i.e.,* $\mathcal{T}(x)$ and $\mathcal{T}(\widehat{x})$) on each learned measurer line, and then use the arc length between nearest points as measured distance results. By the curved measurer lines, our method can measure the intrinsic curvilinear distance more exactly.

proposed a geometric loss to jointly minimize the intra-class distances and maximize the inter-class distances as much as possible. Considering that the above methods utilizing one single matrix $M$ are not flexible for complex data, the traditional Mahalanobis distance is extended to a combined form of multiple linear metrics [30]. Recently, the strategies of adversarial training [8] and collaborative training [20] were introduced in Adversarial Metric Learning (AML) [4] and Bilevel Distance Metric Learning (BDML) [29], respectively, which showed further improvements on the linear metric.

To enhance the flexibility of DML for fitting data pairs from nonlinear sample spaces, the early works transferred the original data points to the high-dimensional kernel space by using traditional kernel methods [26]. Recently, the projection matrix $P$ of the linear DML was extended to a nonlinear feature mapping form $P\mathcal{W}(\cdot)$, in which the mapping $\mathcal{W}(\cdot)$ is implemented by typical Deep Neural Networks (DNN), such as Convolutional Neural Network (CNN) [32] and Multiple Layer Perceptron (MLP) [10]. To further utilize the fitting capability of DNN and characterize the relative distances, the traditional pairwise loss was extended to multi-example forms, such as Npair loss [22] and Angular loss [25]. It is worth pointing out that the above kernelized metrics and DNN based metrics are still calculated with fixed Euclidean distance in the extracted feature space, which ignores the geometric structures of the sample space. To this end, some recent works proposed to learn the projection matrix $P$ on differential manifolds (*e.g.,* SPD manifold [33] and Grassmann manifold [13]) to improve the representation capability on some specific nonlinear data structures. However, the geometries of their used manifolds are usually specified and cannot be learned to adapt to various nonlinear data, and hence remarkably hindering the generality of the manifold based DML approaches.

Although above existing DML models have achieved promising results to some extent, most of them fail to learn the spatial geometric structures of the sample space, and thus their obtained metrics cannot reflect the intrinsic curvilinear distances between data points. To address this challenging problem, in this paper, we firstly present a new interpretation to reveal that the traditional linear distance metric is equivalent to the cumulative arc length between data pair's nearest points on the straight measurer lines (see Fig. 1(a) and (b)). Such straight measurer lines can successfully learn the directions of real distances, but they are not capable of adapting to the curvilinear distance geometries on many nonlinear datasets. Therefore, we propose the Curvilinear Distance Metric Learning (CDML) model, which extends the straight measurer lines to the general smooth curved lines (see Fig. 1(c)). Thanks to the generalized forms of such curvilinear measurers, the geometries of training data can be adaptively learned, so that the nonlinear pairwise distances can be reasonably measured. We theoretically analyze the effectiveness of CDML by showing its fitting capability and generalization bound. Furthermore, we prove that our proposed curvilinear distance satisfies the topological definitions of the (pseudo-)metric, which demonstrates the geometric soundness of such a distance metric. The main contributions of this paper are summarized as: **(I).** We provide a new intuitive interpretation for traditional linear metric learning by explicitly formulating the measurer lines and measurement process; **(II).** We propose a generalized metric learning model dubbed CDML with discovering the curvilinear distance hidden in the nonlinear data, and the corresponding optimization algorithm is designed to solve the proposed model which is guaranteed to converge; **(III).** The complete theoretical guarantee is established, which analyzes the fitting capability, generalization bound, and topological property of CDML, and therefore ensuring the model effectiveness and soundness; **(IV).** CDML is experimentally validated to outperform state-of-the-art metric learning models on both synthetic datasets and real-world datasets.

**Notations.** Throughout this paper, we write matrices, vectors, and 3-order tensors as bold uppercase characters, bold lowercase characters, and bold calligraphic uppercase characters, respectively. For a 3-order tensor $\boldsymbol{\mathcal{M}}$, the notations $\boldsymbol{\mathcal{M}}_{i::}$, $\boldsymbol{\mathcal{M}}_{:i:}$, and $\boldsymbol{\mathcal{M}}_{::i}$ denote the horizontal, lateral, and frontal slices. The tube fiber, row fiber, and column fiber as $\boldsymbol{\mathcal{M}}_{ij:}$, $\boldsymbol{\mathcal{M}}_{i:j}$, and $\boldsymbol{\mathcal{M}}_{:ij}$. Let $\boldsymbol{y} = (y_1, y_2, \cdots, y_N)^\top$ be the label vector of training data pairs $\mathcal{X} = \{(\boldsymbol{x}_j, \widehat{\boldsymbol{x}}_j) | j = 1, 2, \cdots, N\}$ with $\boldsymbol{x}_j, \widehat{\boldsymbol{x}}_j \in \mathbb{R}^d$, where $y_j = 1$ if $\boldsymbol{x}_j$ and $\widehat{\boldsymbol{x}}_j$ are similar, and $y_j = 0$ otherwise. Here $d$ is the data dimensionality, and $N$ is the total number of data pairs. The operators $\|\cdot\|_2$ and $\|\cdot\|_\mathrm{F}$ denote the vector $\ell_2$-norm and matrix (tensor) Frobenius-norm, respectively. The notation $\mathbb{N}_n = \{1, 2, \cdots, n\}$ for any $n \in \mathbb{N}$.

## 2 Curvilinear Distance Metric Learning

In this section, we first present a new geometric interpretation for traditional linear metric learning models. Then the Curvilinear Distance Metric Learning (CDML) is formulated based on such an interpretation. The optimization algorithm is designed to solve CDML with convergence guarantee.

### 2.1 A Geometric Interpretation for Linear Metric

It is well known that the linear distance metric (*i.e.,* squared Mahalanobis distance) [30, 29] between two given data points $\boldsymbol{x}, \widehat{\boldsymbol{x}} \in \mathbb{R}^d$ is defined as

$$\mathrm{Dist}^2_{\boldsymbol{P}}(\boldsymbol{x}, \widehat{\boldsymbol{x}}) = (\boldsymbol{x} - \widehat{\boldsymbol{x}})^\top \boldsymbol{M}(\boldsymbol{x} - \widehat{\boldsymbol{x}}) = (\boldsymbol{x} - \widehat{\boldsymbol{x}})^\top \boldsymbol{P}\boldsymbol{P}^\top(\boldsymbol{x} - \widehat{\boldsymbol{x}}), \tag{1}$$

where the matrix $\boldsymbol{M} = \boldsymbol{P}\boldsymbol{P}^\top$ is assumed to be SPD in $\mathbb{R}^{d \times d}$. In previous works, the above linear distance metric is usually interpreted as the Euclidean distance in the projection space, where the projection matrix $\boldsymbol{P} \in \mathbb{R}^{d \times m}$ plays the role of feature extractions [28]. Here $d$ and $m$ are the dimensionalities of the original sample space and the extracted feature space, respectively. Although such an interpretation offers a friendly way for model extensions, it is not clear enough that why the linear distance metric fails to characterize the curvilinear distances on nonlinear data.

Now we present a new understanding for the linear distance metric from its measurement process, which offers a clear way to hand the nonlinear geometric data. We denote $\boldsymbol{p}_i \in \mathbb{R}^d$ as the $i$-th column of $\boldsymbol{P}$. By using the rule of inner products, Eq. (1) equals to the following cumulative form[1]

$$\sum\nolimits_{i=1}^m \|\boldsymbol{p}_i\|_2^2 \|\boldsymbol{x} - \widehat{\boldsymbol{x}}\|_2^2 \cos^2\langle \boldsymbol{p}_i, \boldsymbol{x} - \widehat{\boldsymbol{x}}\rangle = \sum\nolimits_{i=1}^m \|\boldsymbol{p}_i\|_2^2 \|\boldsymbol{p}_i T_i(\boldsymbol{x}) - \boldsymbol{p}_i T_i(\widehat{\boldsymbol{x}})\|_2^2, \tag{2}$$

where $T_i(\boldsymbol{x})$ and $T_i(\widehat{\boldsymbol{x}})$ are the projection points of $\boldsymbol{x}$ and $\widehat{\boldsymbol{x}}$, which satisfy $(\boldsymbol{p}_i T_i(\boldsymbol{x}) - \boldsymbol{x})^\top \boldsymbol{p}_i = 0$ and $(\boldsymbol{p}_i T_i(\widehat{\boldsymbol{x}}) - \widehat{\boldsymbol{x}})^\top \boldsymbol{p}_i = 0$, respectively. After that, the $\ell_2$-norm distance $\|\boldsymbol{p}_i T_i(\boldsymbol{x}) - \boldsymbol{p}_i T_i(\widehat{\boldsymbol{x}})\|_2$ is equivalently converted to the arc length from $T_i(\boldsymbol{x})$ to $T_i(\widehat{\boldsymbol{x}})$ on the straight line $\boldsymbol{z} = \boldsymbol{p}_i t$ $(t \in \mathbb{R})$, and thus the squared Mahalanobis distance is rewritten as

$$\mathrm{Dist}^2_{\boldsymbol{P}}(\boldsymbol{x}, \widehat{\boldsymbol{x}}) = \sum\nolimits_{i=1}^m \|\boldsymbol{p}_i\|_2^2 \left(\int_{T_i(\boldsymbol{x})}^{T_i(\widehat{\boldsymbol{x}})} \|\boldsymbol{p}_i\|_2 \,\mathrm{d}t\right)^2, \tag{3}$$

where the integral value is the arc length of the straight *measurer line* $\boldsymbol{z} = \boldsymbol{p}_i t$ from $T_i(\boldsymbol{x})$ to $T_i(\widehat{\boldsymbol{x}})$. Here the weight $\|\boldsymbol{p}_i\|_2^2$ is regarded as the scale of the measurer line which equals to the squared unit arc length from 0 to 1. By using the convexity of $g(t) = \|\boldsymbol{p}_i t - \boldsymbol{x}\|_2^2$, the orthogonal condition $(\boldsymbol{p}_i T_i(\boldsymbol{x}) - \boldsymbol{x})^\top \boldsymbol{p}_i = 0$ is equivalent to finding the nearest point $T_i(\boldsymbol{x})$ on the measurer line, namely

$$T_i(\boldsymbol{x}) = \arg\min_{t \in \mathbb{R}} \|\boldsymbol{p}_i t - \boldsymbol{x}\|_2^2. \tag{4}$$

Based on the results of Eq. (3) and Eq. (4), we can clearly observe that the Mahalanobis distance of the data pair $\{\boldsymbol{x}, \widehat{\boldsymbol{x}}\}$ is intrinsically computed as *the cumulative arc length between $\{\boldsymbol{x}, \widehat{\boldsymbol{x}}\}$'s nearest points on the learned measurer line $\boldsymbol{z} = \boldsymbol{p}_i t$*, which is shown in Fig. 1. It reveals that linear metrics merely learn the rough directions of real distances, yet cannot capture the complex data geometry.

### 2.2 Model Establishment

As we mentioned before, traditional metrics learn the direction $\boldsymbol{p}_i$ of the measurer line $\boldsymbol{z} = \boldsymbol{p}_i t$ in the $d$-dimensional sample space. However, such a straight line is far from adapting to complex nonlinear

data in the real world. We thus use a general vector-valued function $\boldsymbol{\theta}_i : \mathbb{R} \to \mathbb{R}^d$ to extend the straight line $\boldsymbol{z} = \boldsymbol{p}_i t$ $(t \in \mathbb{R})$ to *the smooth curved line* $\widetilde{\boldsymbol{z}} = \boldsymbol{\theta}_i(t)$ $(t \in \mathbb{R})$. Specifically, it can be written as

$$\widetilde{\boldsymbol{z}} = (\widetilde{z}_1, \widetilde{z}_2, \cdots, \widetilde{z}_d)^\top = (\theta_{i1}(t), \theta_{i2}(t), \cdots, \theta_{id}(t))^\top = \boldsymbol{\theta}_i(t), \tag{5}$$

where the smooth function $\theta_{ik}(t)$ is the $k$-th element of the vector-valued function $\boldsymbol{\theta}_i(t)$. It should be noticed that such a curved line is also assumed to be zero-crossing, *i.e.*, $\boldsymbol{\theta}_i(0) = \boldsymbol{0}$ which is consistent with the linear distance metric. Then the nearest point $T_i(\boldsymbol{x})$ defined in Eq. (4) can be easily extended to the nearest point set $\mathcal{N}_{\boldsymbol{\theta}_i}(\boldsymbol{x})$, and we naturally have that

$$\mathcal{N}_{\boldsymbol{\theta}_i}(\boldsymbol{x}) = \underset{t \in \mathbb{R}}{\arg\min} \, \|\boldsymbol{\theta}_i(t) - \boldsymbol{x}\|_2^2. \tag{6}$$

Nevertheless, the point set $\mathcal{N}_{\boldsymbol{\theta}_i}(\boldsymbol{x})$ might contain more than one element, so we simply use the smallest element of $\mathcal{N}_{\boldsymbol{\theta}_i}(\boldsymbol{x})$, as described in Definition 1.

**Definition 1.** *For a data point $\boldsymbol{x} \in \mathbb{R}^d$ and a curved line $\boldsymbol{\theta}_i$, we define the calibration point $\mathcal{T}_{\boldsymbol{\theta}_i}(\boldsymbol{x})$ as $\mathcal{T}_{\boldsymbol{\theta}_i}(\boldsymbol{x}) = \arg\min_{t \in \mathcal{N}_{\boldsymbol{\theta}_i}(\boldsymbol{x})} t$, where $\mathcal{N}_{\boldsymbol{\theta}_i}(\boldsymbol{x})$ includes all nearest points of $\boldsymbol{x}$ on the curved line $\boldsymbol{\theta}_i(t)$.*

According to our offered interpretation in Section 2.1, the curvilinear distance is consistently regarded as the cumulative arc length values (see Fig. 1(c)). Here we follow the common formula of arc length in calculus [21], which is given in Definition 2.

**Definition 2.** *The arc length from $T_1$ to $T_2$ on the curved line $\boldsymbol{\theta}_i$ is defined as*

$$\text{Length}_{\boldsymbol{\theta}_i}(T_1, T_2) = \int_{\min(T_1, T_2)}^{\max(T_1, T_2)} \|\boldsymbol{\theta}'_i(t)\|_2 \, \mathrm{d}t, \tag{7}$$

*where the derivative vector $\boldsymbol{\theta}'_i(t) = (\theta'_{i1}(t), \theta'_{i2}(t), \cdots, \theta'_{id}(t))^\top$.*

Based on the above definitions of the arc length and the calibration point, the traditional linear distance metric (*i.e.*, Eq. (3)) is easily extended to the general curvilinear form. Specifically, the squared curvilinear distance between data points $\boldsymbol{x}$ and $\widehat{\boldsymbol{x}}$ is calculated by

$$\text{Dist}_{\boldsymbol{\Theta}}^2(\boldsymbol{x}, \widehat{\boldsymbol{x}}) = \sum_{i=1}^m s_{\boldsymbol{\theta}_i} \cdot \text{Length}_{\boldsymbol{\theta}_i}^2(\mathcal{T}_{\boldsymbol{\theta}_i}(\boldsymbol{x}), \mathcal{T}_{\boldsymbol{\theta}_i}(\widehat{\boldsymbol{x}})), \tag{8}$$

where $\boldsymbol{\Theta} = (\boldsymbol{\theta}_1, \boldsymbol{\theta}_2, \cdots, \boldsymbol{\theta}_m)$ is the learning parameter of the curvilinear distance metric, and $m$ is the number of measurer lines. Here the scale value $s_{\boldsymbol{\theta}_i} = \text{Length}_{\boldsymbol{\theta}_i}^2(0, 1)$. When we use the empirical risk loss to learn $\boldsymbol{\Theta}$, the objective of Curvilinear Distance Metric Learning (CDML) is formulated as

$$\min_{\boldsymbol{\Theta} \in \mathbb{F}_m} \frac{1}{N} \sum_{j=1}^N \mathcal{L}(\text{Dist}_{\boldsymbol{\Theta}}^2(\boldsymbol{x}_j, \widehat{\boldsymbol{x}}_j); y_j) + \lambda \mathcal{R}(\boldsymbol{\Theta}), \tag{9}$$

where $\mathbb{F}_m = \{(\boldsymbol{\theta}_1, \boldsymbol{\theta}_2, \cdots, \boldsymbol{\theta}_m) | \theta_{ik}(t) = 0 \text{ and } \theta_{ik}(t) \text{ is smooth for } i \in \mathbb{N}_m \text{ and } k \in \mathbb{N}_d\}$, the regularization parameter $\lambda > 0$ is tuned by users. In the above objective, the loss function $\mathcal{L}$ evaluates the inconsistency between the predicted distances and their similarity labels, and the regularizer $\mathcal{R}$ is used to reduce the over-fitting.

**Implementation of $\boldsymbol{\Theta}$.** As the learning parameter $\boldsymbol{\Theta}$ of CDML (*i.e.*, Eq. (9)) is in an abstract form and cannot be directly solved, we have to give a concrete form for each curved line $\boldsymbol{\theta}_i(t)$ for learning tasks. Here we employ the polynomial function to approximate $\boldsymbol{\theta}_i(t)$, due to the guarantee of infinite approximation which is described in Theorem 1. It is notable that $\boldsymbol{\theta}_i(t)$ can also be infinitely approximated by other ways, such as Fourier series, deep neural networks, and piecewise linear functions [16]. Without loss of generality, we employ the polynomial function in this paper.

**Theorem 1** (Weierstrass Approximation [21]). *Assume that the vector-valued function $\boldsymbol{\theta}_i(t) \in \mathbb{R}^d$ $(i=1, 2, \cdots, m)$ is continuous and zero-crossing defined on $[a, b]$. Then for any $\epsilon > 0$ and $t \in [a, b]$, there exists the $c$-order polynomial vector-valued function*

$$\boldsymbol{\mathcal{M}}_{i::}(t) = \left( \sum_{k=1}^c \mathcal{M}_{i1k} t^k, \sum_{k=1}^c \mathcal{M}_{i2k} t^k, \cdots, \sum_{k=1}^c \mathcal{M}_{idk} t^k \right)^\top, \tag{10}$$

*such that $\sum_{i=1}^m \|\boldsymbol{\theta}_i(t) - \boldsymbol{\mathcal{M}}_{i::}(t)\|_2 < \epsilon$, where the 3-order tensor $\boldsymbol{\mathcal{M}} = [\mathcal{M}_{ijk}] \in \mathbb{R}^{m \times d \times c}$.*

We let $\boldsymbol{\theta}_i(t) := \boldsymbol{\mathcal{M}}_{i::}(t)$ for $i = 1, 2, \cdots, m$, and then the abstract parameter $\boldsymbol{\Theta}$ in Eq. (9) can be materialized by the tensor $\boldsymbol{\mathcal{M}}$ with infinite approximation in the following optimization objective

$$\min_{\boldsymbol{\mathcal{M}} \in \mathbb{R}^{m \times d \times c}} \frac{1}{N} \sum_{j=1}^N \mathcal{L}(\text{Dist}_{\boldsymbol{\mathcal{M}}}^2(\boldsymbol{x}_j, \widehat{\boldsymbol{x}}_j); y_j) + \lambda \mathcal{R}(\boldsymbol{\mathcal{M}}). \tag{11}$$

We thus successfully convert the abstract optimization problem Eq. (9) to the above concrete form Eq. (11) *w.r.t.* the tensor $\boldsymbol{\mathcal{M}} \in \mathbb{R}^{m \times d \times c}$, which can be easily solved by existing algorithms [11].

**Algorithm 1** Solving Eq. (11) via Stochastic Gradient Descent.

---

**Input:** Training data pairs $\mathcal{X} = \{(\boldsymbol{x}_j, \widehat{\boldsymbol{x}}_j)|j \in \mathbb{N}_N\}$; labels $\boldsymbol{y} \in \{0, 1\}^N$; batch size $h$; learning rate $\rho$; regularization parameter $\lambda$; tensor size parameters $c$ and $m$.

**Initialize:** $k = 1$; $\boldsymbol{\mathcal{M}}^{(1)} = \boldsymbol{0}$.

**Repeat:**

1). Uniformly randomly pick $h$ data pairs $\{(\boldsymbol{x}_{b_j}, \widehat{\boldsymbol{x}}_{b_j})|j \in \mathbb{N}_h\}$ from $\mathcal{X}$.

2). Compute calibration points $\mathcal{T}_{\boldsymbol{\mathcal{M}}_{i::}}(\boldsymbol{x}_{b_j})$ and $\mathcal{T}_{\boldsymbol{\mathcal{M}}_{i::}}(\widehat{\boldsymbol{x}}_{b_j})$ for $i = 1, 2, \cdots, m$ by solving the real roots of $f_i'(t) = 0$ in Eq. (13).

3). Update the learning parameter $\boldsymbol{\mathcal{M}}$ by

$$\boldsymbol{\mathcal{M}}^{(k+1)} := \boldsymbol{\mathcal{M}}^{(k)} - \rho\left(\frac{1}{h}\sum_{j=1}^{h}\mathcal{L}_j'\nabla_{\boldsymbol{\mathcal{M}}^{(k)}}\text{Dist}^2_{\boldsymbol{\mathcal{M}}^{(k)}}(\boldsymbol{x}_{b_j}, \widehat{\boldsymbol{x}}_{b_j}) + \lambda\nabla_{\boldsymbol{\mathcal{M}}^{(k)}}\mathcal{R}(\boldsymbol{\mathcal{M}}^{(k)})\right). \quad (12)$$

4). Update $k := k + 1$.

**Until Converge.**

**Output:** The converged $\boldsymbol{\mathcal{M}}^*$.

---

## 2.3 Optimization Algorithm

Since the pair number $N$ is usually large in Eq. (11), we use the Stochastic Gradient Descent (SGD) method to solve it. As we know that the central operations in SGD are the gradient computation of the objective function. Here we only need to offer the gradient of $\text{Dist}^2_{\boldsymbol{\mathcal{M}}}(\boldsymbol{x}, \widehat{\boldsymbol{x}})$ which mainly depends on the calibration points $\mathcal{T}_{\boldsymbol{\mathcal{M}}_{i::}}(\boldsymbol{x})$, $\mathcal{T}_{\boldsymbol{\mathcal{M}}_{i::}}(\widehat{\boldsymbol{x}})$, and the arc length $\text{Length}_{\boldsymbol{\mathcal{M}}_{i::}}(\mathcal{T}_{\boldsymbol{\mathcal{M}}_{i::}}(\boldsymbol{x}), \mathcal{T}_{\boldsymbol{\mathcal{M}}_{i::}}(\widehat{\boldsymbol{x}}))$.

According to Definition 1, the calibration point $\mathcal{T}_{\boldsymbol{\mathcal{M}}_{i::}}(\boldsymbol{x})$ can be directly obtained from the nearest point set $\mathcal{N}_{\boldsymbol{\mathcal{M}}_{i::}}(\boldsymbol{x}) = \{t^*|f_i(t^*) \leq f_i(\hat{t}), \text{ and } t^*, \hat{t} \in \boldsymbol{\Gamma}_i\}$, where the polynomial function is

$$f_i(t) = \|\boldsymbol{\mathcal{M}}_{i::}(t) - \boldsymbol{x}\|_2^2 = \sum_{j,\,k=1}^{c}(\boldsymbol{\mathcal{M}}_{i:j}^{\top}\boldsymbol{\mathcal{M}}_{i:k})t^{j+k} - 2\sum_{k=1}^{c}\boldsymbol{\mathcal{M}}_{i:k}^{\top}\boldsymbol{x}t^k + \boldsymbol{x}^{\top}\boldsymbol{x}, \quad (13)$$

and $\boldsymbol{\Gamma}_i$ is the real root set for polynomial equation $f_i'(t) = 0$. Here the real roots of $f_i'(t) = 0$ can be efficiently solved by simple numerical algorithms [17], of which the computation complexity does not depend on the number of training data pair $N$ and feature dimensionality $d$.

By using the definition of integral, the arc length is equivalently converted to

$$\text{Length}_{\boldsymbol{\mathcal{M}}_{i::}}(T_1, T_2) = \int_{\min(T_1, T_2)}^{\max(T_1, T_2)}\|\boldsymbol{\mathcal{M}}_{i::}'(t)\|_2 \mathrm{d}t = \lim_{L \to \infty}\sum_{l=0}^{L}\|\boldsymbol{\mathcal{M}}_{i::}'(\min(T_1, T_2) + l\Delta t)\|_2\Delta t, \quad (14)$$

where $\Delta t = |T_1 - T_2|/L$. In practical uses, we fix $L$ to a large number (*e.g.*, $L = 10^3$) and thus obtain a well approximation $G_{\boldsymbol{\mathcal{M}}_{i::}}(T_1, T_2) := (\sum_{l=0}^{L}\|\boldsymbol{\mathcal{M}}_{i::}'(\min(T_1, T_2) + l\Delta t)\|_2\Delta t)^2$ for the squared arc length value $\text{Length}^2_{\boldsymbol{\mathcal{M}}_{i::}}(T_1, T_2)$.

According to the chain rule of derivative, the gradient of $\text{Dist}^2_{\boldsymbol{\mathcal{M}}}$ *w.r.t.* $\boldsymbol{\mathcal{M}}$ is obtained as[2]

$$\nabla_{\boldsymbol{\mathcal{M}}_{i::}}\text{Dist}^2_{\boldsymbol{\mathcal{M}}}(\boldsymbol{x}, \widehat{\boldsymbol{x}}) = \nabla_{\boldsymbol{\mathcal{M}}_{i::}}G_{\boldsymbol{\mathcal{M}}_{i::}}(0, 1) \cdot G_{\boldsymbol{\mathcal{M}}_{i::}}(\mathcal{T}_{\boldsymbol{\mathcal{M}}_{i::}}(\boldsymbol{x}), \mathcal{T}_{\boldsymbol{\mathcal{M}}_{i::}}(\widehat{\boldsymbol{x}}))$$

$$+ G_{\boldsymbol{\mathcal{M}}_{i::}}(0, 1) \cdot \widetilde{\nabla}_{\boldsymbol{\mathcal{M}}_{i::}}G_{\boldsymbol{\mathcal{M}}_{i::}}(\mathcal{T}_{\boldsymbol{\mathcal{M}}_{i::}}(\boldsymbol{x}), \mathcal{T}_{\boldsymbol{\mathcal{M}}_{i::}}(\widehat{\boldsymbol{x}})). \quad (15)$$

It is noticed that the gradient of $G_{\boldsymbol{\mathcal{M}}_{i::}}(\mathcal{T}_{\boldsymbol{\mathcal{M}}_{i::}}(\boldsymbol{x}), \mathcal{T}_{\boldsymbol{\mathcal{M}}_{i::}}(\widehat{\boldsymbol{x}}))$ does not exist necessarily, because the intermediate variable $\mathcal{T}_{\boldsymbol{\mathcal{M}}_{i::}}(\boldsymbol{x})$ is not differentiable *w.r.t.* $\boldsymbol{\mathcal{M}}_{i::}$. Therefore, here we use the smoothed gradient[3] instead of the original gradient of $G_{\boldsymbol{\mathcal{M}}_{i::}}(\mathcal{T}_{\boldsymbol{\mathcal{M}}_{i::}}(\boldsymbol{x}), \mathcal{T}_{\boldsymbol{\mathcal{M}}_{i::}}(\widehat{\boldsymbol{x}}))$. The optimization algorithm for solving Eq. (11) is summarized in Algorithm 1.

**Convergence.** Notice that the main difference between Algorithm 1 and traditional SGD is that we utilize a smoothed gradient instead of the original gradient. Previous works have proved that the smoothed gradient still ensures that the SGD algorithm converges to a stationary point [12, 2].

$G(\mathcal{T}_{\boldsymbol{\mathcal{M}}_{i::}}(\boldsymbol{x}), \mathcal{T}_{\boldsymbol{\mathcal{M}}_{i::}}(\widehat{\boldsymbol{x}})))$, where $\Delta\boldsymbol{\mathcal{M}}_{i::} \in \mathbb{R}^{d \times c}$ is generated from the standard normal distribution, and $\mu > 0$ is a given small number used to smooth the gradient.

# 3 Theoretical Analysis

In this section, we provide theoretical results for the fitting capability, generalization bound, and topological property of CDML. All proofs of theorems are given in the *supplementary materials*.

## 3.1 Fitting Capability

We first reveal that the curvilinear distance learned by Eq. (11) is capable of distinguishing the similarities of all training data pairs. We assume that the (dis)similar pair sets $\mathcal{X}_{\text{Similar}}$ and $\mathcal{X}_{\text{Dissimilar}}$ are the partitions of the whole training data pairs set $\mathcal{X}$, where the similarity label $y_{(\boldsymbol{x}, \widehat{\boldsymbol{x}})} = 1$ if $(\boldsymbol{x}, \widehat{\boldsymbol{x}}) \in \mathcal{X}_{\text{Similar}}$, and $y_{(\boldsymbol{x}, \widehat{\boldsymbol{x}})} = 0$ if $(\boldsymbol{x}, \widehat{\boldsymbol{x}}) \in \mathcal{X}_{\text{Dissimilar}}$. Then the conclusion is described as follows.

**Theorem 2.** *For given $\Delta_{\text{margin}} > 0$, there exist $m, c \in \mathbb{N}$ and $\widetilde{\boldsymbol{\mathcal{M}}} \in \mathbb{R}^{m \times d \times c}$ such that*

$$\text{Dist}_{\widetilde{\boldsymbol{\mathcal{M}}}}(\boldsymbol{\beta}, \widehat{\boldsymbol{\beta}}) - \text{Dist}_{\widetilde{\boldsymbol{\mathcal{M}}}}(\boldsymbol{\alpha}, \widehat{\boldsymbol{\alpha}}) > \Delta_{\text{margin}}, \tag{16}$$

*where $(\boldsymbol{\alpha}, \widehat{\boldsymbol{\alpha}}) \in \mathcal{X}_{\text{Similar}}$ and $(\boldsymbol{\beta}, \widehat{\boldsymbol{\beta}}) \in \mathcal{X}_{\text{Dissimilar}}$.*

From the above result, we know that the well learned curvilinear distance correctly predicts the similarities of data pairs in the training set $\mathcal{X}$, which ensures that the inter-class distances are always greater than the intra-class distances. In most metric learning models, the loss functions are designed to reward the larger inter-class distances and smaller intra-class distances. It means that the distance $\text{Dist}_{\widetilde{\boldsymbol{\mathcal{M}}}}(\cdot, \cdot)$ in Eq. (16) can be successfully achieved by minimizing the loss functions. Therefore, the fitting capability of the curvilinear distance can be reliably guaranteed by the parameter tensor $\boldsymbol{\mathcal{M}}$.

## 3.2 Generalization Bound

Now we further analyze the effectiveness of CDML by offering the generalization bound of the solution to Eq. (11). Such a bound evaluates the bias between the generalization error $\varepsilon(\boldsymbol{\mathcal{M}}) := \mathbb{E}_{(\boldsymbol{x}, \widehat{\boldsymbol{x}}) \sim \mathcal{D}}(\mathcal{L}(\text{Dist}^2_{\boldsymbol{\mathcal{M}}}(\boldsymbol{x}, \widehat{\boldsymbol{x}}); y_{(\boldsymbol{x}, \widehat{\boldsymbol{x}})}))$ and empirical error $\bar{\varepsilon}_{\mathcal{X}}(\boldsymbol{\mathcal{M}}) := \frac{1}{N} \sum_{j=1}^{N} \mathcal{L}(\text{Dist}^2_{\boldsymbol{\mathcal{M}}}(\boldsymbol{x}_j, \widehat{\boldsymbol{x}}_j); y_j)$, where $\mathcal{D}$ is the real data distribution and $\mathbb{E}(\cdot)$ denotes the expectation function. We simply use the squared tensor Frobenius-norm [18] for regularization and have the following conclusion.

**Theorem 3.** *Assume that $\mathcal{R}(\boldsymbol{\mathcal{M}}) = \|\boldsymbol{\mathcal{M}}\|_{\text{F}}^2 = \sum_{i,j,k}(\mathcal{M}_{ijk})^2$ and $\boldsymbol{\mathcal{M}}^* \in \mathbb{R}^{m \times d \times c}$ is the solution to Eq. (11). Then, we have that for any $0 < \delta < 1$ with probability $1 - \delta$*

$$\varepsilon(\boldsymbol{\mathcal{M}}^*) - \bar{\varepsilon}_{\mathcal{X}}(\boldsymbol{\mathcal{M}}^*) \leq X^* \sqrt{2 \ln(1/\delta)/N} + B_\lambda R_N(\mathcal{L}), \tag{17}$$

*where $B_\lambda \to 0$ as $\lambda \to +\infty$[4]. Here $R_N(\mathcal{L})$ is the Rademacher complexity of the loss function $\mathcal{L}$ related to the space $\mathbb{R}^{m \times d \times c}$ for $N$ training pairs, and $X^* = \max_{k \in \mathbb{N}_N} |\mathcal{L}(\text{Dist}^2_{\boldsymbol{\mathcal{M}}^*}(\boldsymbol{x}_k, \widehat{\boldsymbol{x}}_k); y_k)|$.*

In Eq. (17), the first term of the upper bound converges with the increasing of the number of training data pairs $N$. We can also find that the second term converges to 0 with the increasing of $\lambda$, which means the regularizer $\mathcal{R}(\boldsymbol{\mathcal{M}})$ effectively improves the generalization ability of CDML.

## 3.3 Topological Property

In general topology, the *metric*[5] is defined as the distance function satisfying the non-negativity, symmetry, triangle, and coincidence properties [23, 9]. As an extended *metric*, the *pseudo-metric* merely has the first three properties as revealed in [19]. Here we prove that the curvilinear distance defined in Eq. (8) satisfies the topological definitions, and thus its geometric soundness is guaranteed.

**Theorem 4.** *For the curvilinear distance $\text{Dist}_{\boldsymbol{\Theta}}(\boldsymbol{x}, \widehat{\boldsymbol{x}})$ and its corresponding parameter $\boldsymbol{\Theta}$, we denote $\boldsymbol{\Theta}'(\boldsymbol{\tau}) = (\boldsymbol{\theta}'_1(\tau_1), \boldsymbol{\theta}'_2(\tau_2), \cdots, \boldsymbol{\theta}'_m(\tau_m)) \in \mathbb{R}^{d \times m}$ and have that*

*1). $\text{Dist}_{\boldsymbol{\Theta}}(\boldsymbol{x}, \widehat{\boldsymbol{x}})$ is a **pseudo-metric** for any $\boldsymbol{\Theta} \in \mathbb{F}_m$;*

*2). $\text{Dist}_{\boldsymbol{\Theta}}(\boldsymbol{x}, \widehat{\boldsymbol{x}})$ is a **metric**, if $\boldsymbol{\Theta}'(\boldsymbol{\tau})$ is full row rank for any $\boldsymbol{\tau} = (\tau_1, \tau_2, \cdots, \tau_m)^{\top} \in \mathbb{R}^m$.*

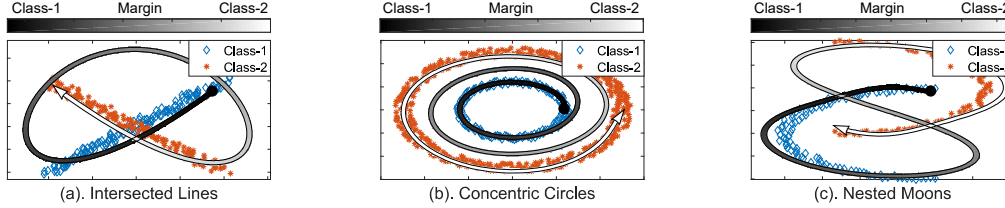

| | Class-1 | Margin | Class-2 | | Class-1 | Margin | Class-2 | | Class-1 | Margin | Class-2 |

(a). Intersected Lines          (b). Concentric Circles          (c). Nested Moons

Figure 2: Visualizations of the measurer lines learned by CDML in two-dimensional space. The grayscale denotes the distance from the origin point to the current point of the learned measurer line.

Table 1: Classification error rates (%, mean $\pm$ std) of all methods on synthetic datasets including *Intersected Lines*, *Concentric Circles*, and *Nested Moons*.

| Datasets | LMNN [27] | ITML [7] | Npair [22] | Angular [25] | ODML [28] | CERML [14] | CDML |
|---|---|---|---|---|---|---|---|
| Instersected Lines | 14.33±1.21 | 17.46±2.11 | 8.52±0.99 | 8.10±3.24 | 10.52±2.17 | 6.21±1.92 | **5.12±1.13** |
| Concentric Circles | 16.62±2.14 | 15.92±3.12 | 9.13±1.51 | 8.98±1.89 | 11.31±2.23 | 10.32±2.16 | **6.95±1.41** |
| Nested Moons | 17.02±2.23 | 12.04±2.14 | 9.22±1.89 | 10.12±2.09 | 15.12±1.98 | 11.12±2.41 | **8.64±2.45** |

Notice that the above Theorem 4 has the same conclusion with the traditional linear distance metric when $\theta_i(t)$ is specialized by $p_i t$, and thus such a result is a generalization of the property in the linear model [23]. In fact, most of the real-world data indeed have non-negativity, symmetry, triangle, and coincidence properties. Hence this theorem clearly tells us that the basic geometric characteristics of real-world data can be well persevered in the curvilinear distance metric.

## 4 Experimental Results

In this section, we show our experimental results on both synthetic and real-world benchmark datasets to validate the effectiveness of CDML. We first visualize the learning results of CDML on synthetic datasets. After that, we compare classification and verification accuracy of CDML with two classical metric learning methods (LMNN [27] and ITML [7]) and four state-of-the-art methods (Npair loss [22], Angular Loss [25], ODML [28], and CERML [14]). Here LMNN, ITML, and ODML are linear and the others are nonlinear. In our experiments, the parameters $\lambda$ and $c$ are fixed to $1.2$ and $10$, respectively. The SGD parameters $h$ and $\rho$ are fixed to $10^3$ and $10^{-3}$, respectively. We follow ITML and use the squared hinge loss and squared Frobenius-norm as $\mathcal{L}$ and $\mathcal{R}$ in Eq. (11), respectively.

### 4.1 Visualizations on Synthetic Datasets

To demonstrate the model effectiveness on nonlinear data, we first visualize the learning results of CDML on nonlinear synthetic datasets including *Intersected Lines*, *Concentric Circles*, and *Nested Moons* [5]. Each dataset contains more than 300 data points across 2 classes in the two-dimensional space. On each dataset, $60\%$ of all data is randomly selected for training, and the rest is used for test. The measurer line count $m$ is fixed to $1$ to clearly visualize the learned results.

As illustrated in Fig. 2, the learned measurer lines are plotted with gray lines, of which the gray-level denotes the arc length distance from the origin point to the current point. According to the definition in Eq. (6), we can clearly observe that the nearest points of data points from two classes are distributed apart on the two sides of the measurer lines (*i.e.,* the low gray-level and high gray-level). Therefore, such measure results correctly predict large values for inter-class distances and small values for intra-class distances. Furthermore, the test error rates of all compared methods are shown in Table 1, and we find that DDML, PML, and CDML obtained superior results over other methods due to their non-linearity. Meanwhile, CDML achieves relatively lower error rates than the deep neural network based model (DDML) and manifold based model (PML) on the three datasets, which validates the superiority of our method.

### 4.2 Comparisons on Classification Datasets

To evaluate the performances of all compared methods on the classification task, we adopt the $k$-NN classifier ($k=5$) based on the learned metrics to investigate the classification error rates of various methods. The datasets are from the well-known UCI machine learning repository [1] including *MNIST, Autompg*, *Sonar*, *Australia, Hayes-r, Glass, Segment, Balance*, *Isolet*, and *Letters*.

Table 2: Classification error rates (%, mean ± std) of all methods on real-world datasets. The last row lists the Win/Tie/Loss counts of CDML against other methods with $t$-test at significance level 0.05.

| Datasets | LMNN [27] | ITML [7] | Npair [22] | Angular [25] | ODML [28] | CERML [14] | CDML |
|---|---|---|---|---|---|---|---|
| MNIST | 17.46±5.32 | 14.32±7.32 | 11.56±1.07 | 12.26±5.12 | 12.12±5.22 | 13.36±2.32 | **8.12±3.64** |
| Autompg | 25.92±3.32 | 26.62±3.21 | 21.95±1.52 | 19.02±3.01 | 25.32±5.32 | 26.36±3.02 | **15.32±6.11** |
| Sonar | 16.04±5.31 | 18.02±3.52 | 15.31±2.56 | 16.86±1.21 | 17.95±6.78 | 19.21±6.33 | 15.40±3.64 |
| Australia | 15.51±2.53 | 17.52±2.13 | 15.12±5.11 | 15.54±1.23 | 16.23±4.12 | 18.26±6.22 | **12.22±2.54** |
| Hayes-r | 30.46±7.32 | 34.24±6.32 | 24.36±2.17 | **23.12±1.37** | 29.76±1.07 | 30.12±5.32 | 25.15±5.23 |
| Glass | 30.12±2.32 | 29.11±3.28 | 22.32±4.72 | 23.02±1.22 | 28.26±1.22 | 29.11±0.12 | **22.12±4.64** |
| Segment | 2.73±0.82 | 5.16±2.22 | 8.77±0.32 | 4.11±1.22 | 3.76±1.34 | 5.36±3.12 | **1.23±0.32** |
| Balance | 9.93±1.62 | 9.31±2.21 | 8.12±1.97 | 7.12±2.22 | 8.63±2.22 | 9.45±5.45 | **5.01±2.64** |
| IsoLet | 3.23±1.32 | 9.23±2.32 | 5.43±2.12 | 5.49±1.12 | **2.68±0.72** | 7.26±2.32 | 3.12±1.64 |
| Letters | 4.21±2.05 | 6.24±0.32 | 5.14±1.04 | 4.67±1.82 | 4.88±0.82 | 5.32±2.22 | **2.09±0.64** |
| W/T/L | 8/2/0 | 9/1/0 | 5/5/0 | 6/4/0 | 8/2/0 | 8/2/0 | — |

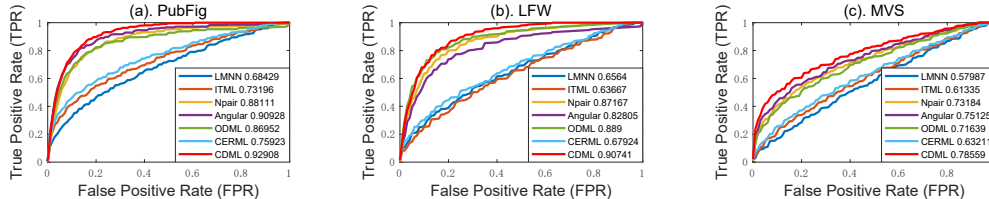

Figure 3: ROC curves of all methods on the 3 datasets. AUC values are presented in the legends.

We compare all methods over 20 random trials. In each trial, 80% of examples are randomly selected as the training examples, and the rest are used for testing. The training pairs are generated by randomly picking up $1000K(K-1)$ pairs among the training examples [31], where $K$ is the number of classes. Here the measurer line count $m$ is fixed to the feature dimensionality (*i.e., d*). The average classification error rates of all compared methods are shown in Table 2. We also perform the $t$-test (significance level 0.05) to validate the superiority of our method over all baseline methods on each dataset. From the experimental results, we can observe that CDML obtains significant improvements on the linear metric learning models, which demonstrates the usefulness of our proposed curvilinear generalization. Furthermore, the statistical records of average error rates and $t$-test results reliably validate the superiority of our method over other baseline methods.

### 4.3 Comparisons on Verification Datasets

We use two face datasets and one image matching dataset to evaluate the capabilities of all compared methods on image verification. The *PubFig* face dataset includes $2 \times 10^4$ image pairs belonging to 140 people [15], in which the first 80% data pairs are selected for training and the rest are used for test. Similar experiments are performed on the *LFW* face dataset which includes 13233 unconstrained face images [15]. The *MVS* dataset [3] consists of over $3 \times 10^4$ image patches sampled from 3D objects, in which $10^5$ pairs are selected to form the training set, and $10^4$ pairs are used for test.

The adopted features are extracted by DSIFT [6] and Siamese-CNN [32] for face datasets and image patch dataset, respectively. We plot the Receiver Operator Characteristic (ROC) curve by changing the thresholds of different distance metrics. Then the values of Area Under Curve (AUC) are calculated to quantitatively evaluate the performances of all comparators. From the ROC curves and AUC values in Fig. 3, it is clear to see that DDML and CDML consistently outperform other methods. In comparison, CDML obtains better results than the best baseline method DDML on three datasets.

## 5 Conclusion

In this paper, we introduced the new insight of the mechanism of metric learning models, where the measured distance is naturally interpreted as the arc length between nearest points on the learned straight measurer lines. We extended such straight measurer lines to general curved lines for further learning the intrinsic geometries of the training data. To the best of our knowledge, this is the first work of metric learning with adaptively constructing the geometric relations between data points. We provided theoretical analysis to show that the proposed framework can be well applied to the general nonlinear data. Visualizations on toy data indicate that the learned measurer lines critically capture the underlying rules, and thus making the learning algorithm acquire more reliable and precise metric than the state-of-the-art methods.

**Acknowledgment**

S.C., J.Y., and C.G. were supported by the National Science Fund (NSF) of China under Grant (Nos: U1713208, 61602246, and 61973162), Program for Changjiang Scholars, "111" Program AH92005, the Fundamental Research Funds for the Central Universities (No: 30918011319), NSF of Jiangsu Province (No: BK20171430), the "Young Elite Scientists Sponsorship Program" by Jiangsu Province, and the "Young Elite Scientists Sponsorship Program" by CAST (No: 2018QNRC001).

L.L. and H.H. were supported by U.S. NSF-IIS 1836945, NSF-IIS 1836938, NSF-DBI 1836866, NSF-IIS 1845666, NSF-IIS 1852606, NSF-IIS 1838627, and NSF-IIS 1837956.

## Footnotes

[1]For calculation details, $\mathrm{Dist}^2_{\boldsymbol{P}}(\boldsymbol{x}, \widehat{\boldsymbol{x}}) = (\boldsymbol{x} - \widehat{\boldsymbol{x}})^\top \boldsymbol{P}\boldsymbol{P}^\top(\boldsymbol{x} - \widehat{\boldsymbol{x}}) = \|\boldsymbol{P}^\top(\boldsymbol{x} - \widehat{\boldsymbol{x}})\|_2^2 = \|(\boldsymbol{p}_1^\top(\boldsymbol{x} - \widehat{\boldsymbol{x}}), \boldsymbol{p}_2^\top(\boldsymbol{x} - \widehat{\boldsymbol{x}}), \cdots, \boldsymbol{p}_m^\top(\boldsymbol{x} - \widehat{\boldsymbol{x}}))\|_2^2 = \sum_{i=1}^m (\boldsymbol{p}_i^\top(\boldsymbol{x} - \widehat{\boldsymbol{x}}))^2 = \sum_{i=1}^m \|\boldsymbol{p}_i\|_2^2 \|\boldsymbol{x} - \widehat{\boldsymbol{x}}\|_2^2 \cos^2\langle \boldsymbol{p}_i, \boldsymbol{x} - \widehat{\boldsymbol{x}}\rangle$.

[2]$\nabla_{\boldsymbol{\mathcal{M}}_{i::}}G_{\boldsymbol{\mathcal{M}}_{i::}}(0, 1) = \frac{2}{L}\sqrt{G_{\boldsymbol{\mathcal{M}}_{i::}}(0, 1)}\sum_{l=0}^{L}\frac{\boldsymbol{\mathcal{M}}_{i::}'(l/L)\cdot((l/L)^1, (l/L)^2, \cdots, (l/L)^c)}{\|\boldsymbol{\mathcal{M}}_{i::}'(l/L)\|_2}.$

[3]$\widetilde{\nabla}_{\boldsymbol{\mathcal{M}}_{i::}}G_{\boldsymbol{\mathcal{M}}_{i::}}(\mathcal{T}_{\boldsymbol{\mathcal{M}}_{i::}}(\boldsymbol{x}), \mathcal{T}_{\boldsymbol{\mathcal{M}}_{i::}}(\widehat{\boldsymbol{x}})) = \frac{\Delta\boldsymbol{\mathcal{M}}_{i::}}{\mu\|\Delta\boldsymbol{\mathcal{M}}_{i::}\|_2^2}(G_{\boldsymbol{\mathcal{M}}_{i::}+\mu\Delta\boldsymbol{\mathcal{M}}_{i::}}(\mathcal{T}_{\boldsymbol{\mathcal{M}}_{i::}+\mu\Delta\boldsymbol{\mathcal{M}}_{i::}}(\boldsymbol{x}), \mathcal{T}_{\boldsymbol{\mathcal{M}}_{i::}+\mu\Delta\boldsymbol{\mathcal{M}}_{i::}}(\widehat{\boldsymbol{x}})) -$

[4]Here $B_\lambda = 2\mathbb{E}_{\mathcal{X}, \mathcal{Z}}[\sup_{\boldsymbol{\mathcal{M}} \in \mathcal{F}(\lambda)} \bar{\varepsilon}_{\mathcal{Z}}(\boldsymbol{\mathcal{M}}) - \bar{\varepsilon}_{\mathcal{X}}(\boldsymbol{\mathcal{M}})] / \mathbb{E}_{\mathcal{X}, \mathcal{Z}}[\sup_{\boldsymbol{\mathcal{M}} \in \mathbb{R}^{m \times d \times c}} \bar{\varepsilon}_{\mathcal{Z}}(\boldsymbol{\mathcal{M}}) - \bar{\varepsilon}_{\mathcal{X}}(\boldsymbol{\mathcal{M}})]$ and $\mathcal{F}(\lambda)$ is a shrinking hypothesis space induced by the regularizer $\mathcal{R}(\boldsymbol{\mathcal{M}})$.

[5]The distance function $\text{Dist}(\cdot, \cdot)$ is a metric if and only if it satisfies the four conditions for any $\boldsymbol{\alpha}_1, \boldsymbol{\alpha}_2, \boldsymbol{\alpha}_3 \in \mathbb{R}^d$: **(I).** Non-negativity: $\text{Dist}(\boldsymbol{\alpha}_1, \boldsymbol{\alpha}_2) \geq 0$; **(II).** Symmetry: $\text{Dist}(\boldsymbol{\alpha}_1, \boldsymbol{\alpha}_2) = \text{Dist}(\boldsymbol{\alpha}_2, \boldsymbol{\alpha}_1)$; **(III).** Triangle: $\text{Dist}(\boldsymbol{\alpha}_1, \boldsymbol{\alpha}_2) + \text{Dist}(\boldsymbol{\alpha}_2, \boldsymbol{\alpha}_3) \geq \text{Dist}(\boldsymbol{\alpha}_1, \boldsymbol{\alpha}_3)$; **(IV).** Coincidence: $\text{Dist}(\boldsymbol{\alpha}_1, \boldsymbol{\alpha}_2) = 0 \Longleftrightarrow \boldsymbol{\alpha}_1 = \boldsymbol{\alpha}_2$.

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
