[Supplementary Material]

# Supplementary Material for "Curvilinear Distance Metric Learning"

## Abstract

This supplementary document contains all technical proofs for **Theorem 2**, **Theorem 3**, and **Theorem 4** in the NeurIPS_2019 paper entitled "Curvilinear Distance Metric Learning". It is indeed the appendix section of the paper.

## A   Proof of Theorem 2 (Fitting Capability)

We introduce the following Lemma 1 for proving our Theorem 2.

**Lemma 1.** *Assume that $s_1, s_2, \cdots, s_K > 0$ and $t_j = j + \kappa_{\boldsymbol{s}}(j)\Delta$ for $j = 1, 2, \cdots, H$, where $H = \sum_{i=1}^{K} s_i$ and $\kappa_{\boldsymbol{s}}(j)$ denotes the maximal integer satisfying $\sum_{i=1}^{\kappa_{\boldsymbol{s}}(j)} s_i < j$. Then for the Vandermonde matrix*

$$\boldsymbol{V}_{t_1, t_2, \cdots, t_H}(\Delta) = \begin{pmatrix} 1 & 0 & \cdots & 0 \\ 1 & t_1 & \cdots & t_1^H \\ \vdots & \vdots & \ddots & \vdots \\ 1 & t_H & \cdots & t_H^H \end{pmatrix} \in \mathbb{R}^{(H+1) \times (H+1)}, \tag{A.1}$$

*the limitation of $(\boldsymbol{V}_{t_1, t_2, \cdots, t_H}(\Delta))^{-1}$ exists as $\Delta \to +\infty$.*

*Proof.* As $1 = t_1 < t_2 < \cdots < t_H$, it holds that

$$\det(\boldsymbol{V}_{t_1, t_2, \cdots, t_H}(\Delta)) = \prod_{1 \leq i < j \leq H}(t_j - t_i) \neq 0, \tag{A.2}$$

which implies that the matrix $\boldsymbol{V}_{t_1, t_2, \cdots, t_H}(\Delta)$ is invertible. Then we denote the *adjoint matrix* of $\boldsymbol{V}_{t_1, t_2, \cdots, t_H}(\Delta)$ as $\boldsymbol{V}^{\star} \in \mathbb{R}^{(H+1) \times (H+1)}$, and we have

$$(\boldsymbol{V}_{t_1, t_2, \cdots, t_H}(\Delta))^{-1} = \frac{\boldsymbol{V}^{\star}}{\det(\boldsymbol{V}_{t_1, t_2, \cdots, t_H}(\Delta))}, \tag{A.3}$$

where $V_{ij}^{\star} = (-1)^{i+j} V(i,j)$ and $V(i,j) \in \mathbb{R}$ is the cofactor of $\boldsymbol{V}_{t_1, t_2, \cdots, t_H}(\Delta)$ *w.r.t.* $i$-th row and $j$-column. According to the definition of determinant, $\det(\boldsymbol{V}_{t_1, t_2, \cdots, t_H}(\Delta))$ can be written as

$$\det(\boldsymbol{V}_{t_1, t_2, \cdots, t_H}(\Delta)) = \sum_{k=0}^{Q} u_k \Delta^k, \tag{A.4}$$

where the polynomial order $Q \leq H$, and the polynomial coefficients $\boldsymbol{u} = (u_0, u_1, \cdots, u_Q)^{\top}$. For the cofactor $V(i,j)$, we have that

$$V(i,j) = \sum_{k=0}^{P^{(i,j)}} v_k^{(i,j)} \Delta^k, \tag{A.5}$$

and the polynomial order $P^{(i,j)} \leq Q$ can be directly obtained from the definition of the cofactor, in which the polynomial coefficients $\boldsymbol{v}^{(i,j)} = (v_0^{(i,j)}, v_1^{(i,j)}, \cdots, v_{P^{(i,j)}}^{(i,j)})^{\top}$. Then for Eq. (A.3), we have

$$\lim_{\Delta \to +\infty} \left| \frac{V_{ij}^{\star}}{\det(\boldsymbol{V}_{t_1, t_2, \cdots, t_H}(\Delta))} \right| = \lim_{\Delta \to +\infty} \frac{\left| \sum_{k=0}^{P^{(i,j)}} v_k^{(i,j)} \Delta^k \right|}{\left| \sum_{k=0}^{Q} u_k \Delta^k \right|} = \lim_{\Delta \to +\infty} \left| \frac{v_{P^{(i,j)}}^{(i,j)} \Delta^{P^{(i,j)}}}{u_Q \Delta^Q} \right|, \tag{A.6}$$

and

$$\lim_{\Delta \to +\infty} \left| \frac{v_{P^{(i,j)}}^{(i,j)} \Delta^{P^{(i,j)}}}{u_Q \Delta^Q} \right| = \begin{cases} 0, & \text{if } P^{(i,j)} < Q, \\ \left| \frac{v_{P^{(i,j)}}^{(i,j)}}{u_Q} \right|, & \text{if } P^{(i,j)} = Q. \end{cases} \tag{A.7}$$

where $i, j \in \mathbb{N}_{H+1}$. Therefore, the limitation of $(\boldsymbol{V}_{t_1, t_2, \cdots, t_H}(\Delta))^{-1}$ exists as $\Delta \to +\infty$. □

**Theorem 2.** *For given* $\Delta_{\mathrm{margin}} > 0$, *there exist* $m, c \in \mathbb{N}$ *and* $\widetilde{\mathcal{M}} \in \mathbb{R}^{m \times d \times c}$ *such that*

$$\mathrm{Dist}_{\widetilde{\mathcal{M}}}(\boldsymbol{\beta}, \widehat{\boldsymbol{\beta}}) - \mathrm{Dist}_{\widetilde{\mathcal{M}}}(\boldsymbol{\alpha}, \widehat{\boldsymbol{\alpha}}) > \Delta_{\mathrm{margin}}, \tag{A.8}$$

*where* $(\boldsymbol{\alpha}, \widehat{\boldsymbol{\alpha}}) \in \mathcal{X}_{\mathrm{Similar}}$ *and* $(\boldsymbol{\beta}, \widehat{\boldsymbol{\beta}}) \in \mathcal{X}_{\mathrm{Dissimilar}}$.

*Proof.* We first convert the point pair sets $\mathcal{X}_{\mathrm{Similar}}$ and $\mathcal{X}_{\mathrm{Dissimilar}}$ to point sets $A_1, A_2, \cdots, A_K$ of $K$ categories. Specifically, the pair sets can be written as

$$\begin{cases} \mathcal{X}_{\mathrm{Similar}} & = \cup_{i=1}^K (A_i \times A_i), \\ \mathcal{X}_{\mathrm{Dissimilar}} & = \cup_{i \neq j} (A_i \times A_j), \end{cases} \tag{A.9}$$

where "$\times$" denotes the *Cartesian Product* [2] of two sets. Assume that

$$\begin{cases} A_1 = \{\boldsymbol{a}_1^{(1)}, \boldsymbol{a}_1^{(2)}, \cdots, \boldsymbol{a}_1^{(|A_1|)} \in \mathbb{R}^d\}, \\ A_2 = \{\boldsymbol{a}_2^{(1)}, \boldsymbol{a}_2^{(2)}, \cdots, \boldsymbol{a}_2^{(|A_2|)} \in \mathbb{R}^d\}, \\ \cdots, \\ A_K = \{\boldsymbol{a}_K^{(1)}, \boldsymbol{a}_K^{(2)}, \cdots, \boldsymbol{a}_K^{(|A_K|)} \in \mathbb{R}^d\}. \end{cases} \tag{A.10}$$

where $|A_i|$ denotes the cardinality of the set $A_i$ for $i = 1, 2, \cdots, K$. Let $t_j = j + \kappa(j)\Delta$ and[1]

$$(\boldsymbol{b}_1, \cdots, \boldsymbol{b}_H) = (\boldsymbol{a}_1^{(1)}, \cdots, \boldsymbol{a}_1^{(|A_1|)}, \boldsymbol{a}_2^{(1)}, \cdots, \boldsymbol{a}_2^{(|A_2|)}, \cdots, \boldsymbol{a}_K^{(1)}, \cdots, \boldsymbol{a}_K^{(|A_K|)}), \tag{A.11}$$

where $\boldsymbol{b}_j \in \mathbb{R}^d$, $j \in \mathbb{N}_H$, $\Delta > 0$, $H = \sum_{i=1}^K |A_i|$. We further denote that $t_0 = 0$, $\boldsymbol{b}_0 = \boldsymbol{0} \in \mathbb{R}^d$ and construct the following *Vandermonde matrix*

$$\boldsymbol{V}_{t_1, t_2, \cdots, t_H}(\Delta) = \begin{pmatrix} 1 & t_0 & \cdots & t_0^H \\ 1 & t_1 & \cdots & t_1^H \\ \vdots & \vdots & \ddots & \vdots \\ 1 & t_H & \cdots & t_H^H \end{pmatrix} = \begin{pmatrix} 1 & 0 & \cdots & 0 \\ 1 & t_1 & \cdots & t_1^H \\ \vdots & \vdots & \ddots & \vdots \\ 1 & t_H & \cdots & t_H^H \end{pmatrix} \in \mathbb{R}^{(H+1) \times (H+1)}. \tag{A.12}$$

Then we have

$$\det(\boldsymbol{V}_{t_1, t_2, \cdots, t_H}(\Delta)) = \prod_{1 \leq i < j \leq H} (t_j - t_i) \neq 0. \tag{A.13}$$

Therefore, the equation group $\boldsymbol{V}_{t_1, t_2, \cdots, t_H}(\Delta)\boldsymbol{\mu}_k = (b_{0k}, b_{1k}, \cdots, b_{Hk})^\top$ has the unique solution $\boldsymbol{\mu}_k = (\mu_{k0}, \mu_{k1}, \cdots, \mu_{kH})^\top$, in which $b_{jk}$ denotes the $k$-th element of the vector $\boldsymbol{b}_j$ and $k = 1, 2, \cdots, d$. It implies that the polynomial function $f_{\boldsymbol{\mu}_k}(t) = \sum_{i=0}^H \mu_{ki} t^i$ crosses the points $(t_0, b_{0k}), (t_1, b_{1k}), \cdots, (t_H, b_{Hk})$ successively for $k = 1, 2, \cdots, d$.

Without loss of generality, we assume that there exist real numbers $\widetilde{t}_1, \widetilde{t}_2, \cdots, \widetilde{t}_S \notin \{t_0, t_1, \cdots, t_H\}$ and function $l(j)$ such that

$$(f_{\boldsymbol{\mu}_1}(\widetilde{t}_j), f_{\boldsymbol{\mu}_2}(\widetilde{t}_j), \cdots, f_{\boldsymbol{\mu}_{d-1}}(\widetilde{t}_j)) = (b_{l(j)1}, b_{l(j)2}, \cdots, b_{l(j)(d-1)}), \tag{A.14}$$

where $j \in \mathbb{N}_S$. Since the $H$-order polynomial equation exists $H$ real roots at most, we can easily obtain $S \leq H(H-1)$. Then we assume that $\{\widetilde{t}_1, \widetilde{t}_2, \cdots, \widetilde{t}_S\} = \{\widetilde{t}_1^+, \widetilde{t}_2^+, \cdots, \widetilde{t}_U^+\} \cup \{\widetilde{t}_1^-, \widetilde{t}_2^-, \cdots, \widetilde{t}_V^-\}$ which satisfies

$$\begin{cases} f_{\boldsymbol{\mu}_d}(\widetilde{t}_j^+) = b_{l(j)d}, & \text{for } j = 1, 2, \cdots, U, \\ f_{\boldsymbol{\mu}_d}(\widetilde{t}_j^-) \neq b_{l(j)d}, & \text{for } j = 1, 2, \cdots, V. \end{cases} \tag{A.15}$$

We construct the following function

$$\widetilde{f}_{\boldsymbol{\mu}_d}(t) = f_{\boldsymbol{\mu}_d}(t) + \frac{1}{\prod_{i=0}^{H}(|t_i|+1)\prod_{j=1}^{V}(|\widetilde{t}_j^-|+1)}\prod_{i=0}^{H}(t-t_i)\prod_{j=1}^{V}(t-\widetilde{t}_j^-), \qquad (A.16)$$

which satisfies

$$\begin{cases} \widetilde{f}_{\boldsymbol{\mu}_d}(t) = f_{\boldsymbol{\mu}_d}(t), & t \in \{t_0, t_1, \cdots, t_H\} \cup \{\widetilde{t}_1^-, \widetilde{t}_2^-, \cdots, \widetilde{t}_V^-\}, \\ \widetilde{f}_{\boldsymbol{\mu}_d}(t) \neq f_{\boldsymbol{\mu}_d}(t), & t \notin \{t_0, t_1, \cdots, t_H\} \cup \{\widetilde{t}_1^-, \widetilde{t}_2^-, \cdots, \widetilde{t}_V^-\}. \end{cases} \qquad (A.17)$$

It is easy to verify that for any $j \in \mathbb{N}_S$, we have

$$\widetilde{f}_{\boldsymbol{\mu}_d}(\widetilde{t}_j) \neq b_{l(j)d}. \qquad (A.18)$$

According to Eq. (A.14) and Eq. (A.18), it follows that for $t \in \{\widetilde{t}_1, \widetilde{t}_2, \cdots, \widetilde{t}_S\}^2$

$$(f_{\boldsymbol{\mu}_1}(t), f_{\boldsymbol{\mu}_2}(t), \cdots, f_{\boldsymbol{\mu}_d}(t), \widetilde{f}_{\boldsymbol{\mu}_d}(t))^\top \notin \{\boldsymbol{b}_0, \boldsymbol{b}_1, \cdots, \boldsymbol{b}_H\}. \qquad (A.19)$$

Furthermore, for $t \in \mathbb{R}\backslash\{\widetilde{t}_1, \widetilde{t}_2, \cdots, \widetilde{t}_S, t_0, t_1, \cdots, t_H\}$, it holds that[3]

$$(f_{\boldsymbol{\mu}_1}(t), f_{\boldsymbol{\mu}_2}(t), \cdots, f_{\boldsymbol{\mu}_d}(t), \widetilde{f}_{\boldsymbol{\mu}_d}(t))^\top \notin \{\boldsymbol{b}_0, \boldsymbol{b}_1, \cdots, \boldsymbol{b}_H\}. \qquad (A.20)$$

In summary, for any $t \in \mathbb{R}$, we have

$$\begin{cases} (f_{\boldsymbol{\mu}_1}(t), f_{\boldsymbol{\mu}_2}(t), \cdots, f_{\boldsymbol{\mu}_d}(t), \widetilde{f}_{\boldsymbol{\mu}_d}(t))^\top \in \{\boldsymbol{b}_0, \boldsymbol{b}_1, \cdots, \boldsymbol{b}_H\}, & \text{if } t \in \{t_0, t_1, \cdots, t_H\}, \\ (f_{\boldsymbol{\mu}_1}(t), f_{\boldsymbol{\mu}_2}(t), \cdots, f_{\boldsymbol{\mu}_d}(t), \widetilde{f}_{\boldsymbol{\mu}_d}(t))^\top \notin \{\boldsymbol{b}_0, \boldsymbol{b}_1, \cdots, \boldsymbol{b}_H\}, & \text{if } t \notin \{t_0, t_1, \cdots, t_H\}. \end{cases} \qquad (A.21)$$

Namely we have that $(f_{\boldsymbol{\mu}_1}(t), f_{\boldsymbol{\mu}_2}(t), \cdots, f_{\boldsymbol{\mu}_d}(t), \widetilde{f}_{\boldsymbol{\mu}_d}(t))^\top \in \{\boldsymbol{b}_0, \boldsymbol{b}_1, \cdots, \boldsymbol{b}_H\}$ if and only if $t \in \{t_0, t_1, \cdots, t_H\}$. Let

$$\boldsymbol{\omega}(t) = (f_{\boldsymbol{\mu}_1}(t), f_{\boldsymbol{\mu}_2}(t), \cdots, f_{\boldsymbol{\mu}_d}(t), \widetilde{f}_{\boldsymbol{\mu}_d}(t))^\top, \qquad (A.22)$$

then we thus have that $\boldsymbol{\omega}(t)$ is invertible at $t = t_0, t_1, \cdots, t_H$, *i.e.*, $t_i = \boldsymbol{\omega}^{-1}(\boldsymbol{b}_i)$ for $i = 0, 1, \cdots, H$. We denote $D^+$ and $D^-$ as

$$\begin{cases} D^+ = \max_{(\boldsymbol{\alpha}, \widehat{\boldsymbol{\alpha}}) \in \mathcal{X}_{\text{Similar}}} \text{Length}_{\boldsymbol{\omega}}(\boldsymbol{\alpha}, \widehat{\boldsymbol{\alpha}}), \\ D^- = \min_{(\boldsymbol{\beta}, \widehat{\boldsymbol{\beta}}) \in \mathcal{X}_{\text{Dissimilar}}} \text{Length}_{\boldsymbol{\omega}}(\boldsymbol{\beta}, \widehat{\boldsymbol{\beta}}), \end{cases} \qquad (A.23)$$

then we have

$$D^+ = \max_{(\boldsymbol{\alpha}, \widehat{\boldsymbol{\alpha}}) \in \mathcal{X}_{\text{Similar}}} \int_{\min(\mathcal{T}_{\boldsymbol{\omega}}(\boldsymbol{\alpha}), \mathcal{T}_{\boldsymbol{\omega}}(\widehat{\boldsymbol{\alpha}}))}^{\max(\mathcal{T}_{\boldsymbol{\omega}}(\boldsymbol{\alpha}), \mathcal{T}_{\boldsymbol{\omega}}(\widehat{\boldsymbol{\alpha}}))} \|\boldsymbol{\omega}'(t)\|_2 dt = \max_{(\boldsymbol{\alpha}, \widehat{\boldsymbol{\alpha}}) \in \mathcal{X}_{\text{Similar}}} \int_{\min(\boldsymbol{\omega}^{-1}(\boldsymbol{\alpha}), \boldsymbol{\omega}^{-1}(\widehat{\boldsymbol{\alpha}}))}^{\max(\boldsymbol{\omega}^{-1}(\boldsymbol{\alpha}), \boldsymbol{\omega}^{-1}(\widehat{\boldsymbol{\alpha}}))} \|\boldsymbol{\omega}'(t)\|_2 dt, \qquad (A.24)$$

and

$$D^- = \min_{(\boldsymbol{\beta}, \widehat{\boldsymbol{\beta}}) \in \mathcal{X}_{\text{Dissimilar}}} \int_{\min(\mathcal{T}_{\boldsymbol{\omega}}(\boldsymbol{\beta}), \mathcal{T}_{\boldsymbol{\omega}}(\widehat{\boldsymbol{\beta}}))}^{\min(\mathcal{T}_{\boldsymbol{\omega}}(\boldsymbol{\beta}), \mathcal{T}_{\boldsymbol{\omega}}(\widehat{\boldsymbol{\beta}}))} \|\boldsymbol{\omega}'(t)\|_2 dt = \min_{(\boldsymbol{\beta}, \widehat{\boldsymbol{\beta}}) \in \mathcal{X}_{\text{Dissimilar}}} \int_{\min(\boldsymbol{\omega}^{-1}(\boldsymbol{\beta}), \boldsymbol{\omega}^{-1}(\widehat{\boldsymbol{\beta}}))}^{\max(\boldsymbol{\omega}^{-1}(\boldsymbol{\beta}), \boldsymbol{\omega}^{-1}(\widehat{\boldsymbol{\beta}}))} \|\boldsymbol{\omega}'(t)\|_2 dt. \qquad (A.25)$$

By Lemma 1, it follows that for $k \in \mathbb{N}_d$

$$\lim_{\Delta \to +\infty} \boldsymbol{\mu}_k = (\boldsymbol{V}_{t_1, t_2, \cdots, t_H}(\Delta))^{-1}(b_{0k}, b_{1k}, \cdots, b_{Hk})^\top = \boldsymbol{\mu}_k^*. \qquad (A.26)$$

According to Eq. (A.22), it follows that the coefficients of the polynomial function $g(t) = \|\boldsymbol{\omega}'(t)\|_2^2$ converge as $\Delta \to +\infty$. Then we have

$$\lim_{\Delta \to +\infty} \frac{D^+}{D^-}$$

$$= \lim_{\Delta \to +\infty} \frac{\max\limits_{(\boldsymbol{\alpha},\widehat{\boldsymbol{\alpha}}) \in \mathcal{X}_{\text{Similar}}} \int_{\min(\boldsymbol{\omega}^{-1}(\boldsymbol{\alpha}),\boldsymbol{\omega}^{-1}(\widehat{\boldsymbol{\alpha}}))}^{\max(\boldsymbol{\omega}^{-1}(\boldsymbol{\alpha}),\boldsymbol{\omega}^{-1}(\widehat{\boldsymbol{\alpha}}))} \sqrt{g(t)}\mathrm{d}t}{\min\limits_{(\boldsymbol{\beta},\widehat{\boldsymbol{\beta}}) \in \mathcal{X}_{\text{Dissimilar}}} \int_{\min(\boldsymbol{\omega}^{-1}(\boldsymbol{\beta}),\boldsymbol{\omega}^{-1}(\widehat{\boldsymbol{\beta}}))}^{\max(\boldsymbol{\omega}^{-1}(\boldsymbol{\beta}),\boldsymbol{\omega}^{-1}(\widehat{\boldsymbol{\beta}}))} \sqrt{g(t)}\mathrm{d}t}$$

$$\leq \lim_{\Delta \to +\infty} \frac{\max\limits_{(\boldsymbol{\alpha},\widehat{\boldsymbol{\alpha}}) \in \mathcal{X}_{\text{Similar}}} \int_{\min(\boldsymbol{\omega}^{-1}(\boldsymbol{\alpha}),\boldsymbol{\omega}^{-1}(\widehat{\boldsymbol{\alpha}}))}^{\max(\boldsymbol{\omega}^{-1}(\boldsymbol{\alpha}),\boldsymbol{\omega}^{-1}(\widehat{\boldsymbol{\alpha}}))} \sqrt{g(t)}\mathrm{d}t}{\min\limits_{(\boldsymbol{\beta},\widehat{\boldsymbol{\beta}}) \in \mathcal{X}_{\text{Dissimilar}}} \int_{\min(\boldsymbol{\omega}^{-1}(\boldsymbol{\beta}),\boldsymbol{\omega}^{-1}(\widehat{\boldsymbol{\beta}}))+\frac{1}{2}|\boldsymbol{\omega}^{-1}(\boldsymbol{\beta})-\boldsymbol{\omega}^{-1}(\widehat{\boldsymbol{\beta}})|}^{\max(\boldsymbol{\omega}^{-1}(\boldsymbol{\beta}),\boldsymbol{\omega}^{-1}(\widehat{\boldsymbol{\beta}}))} \sqrt{g(t)}\mathrm{d}t}$$

$$\leq \lim_{\Delta \to +\infty} \frac{\left(\max\limits_{(\boldsymbol{\alpha},\widehat{\boldsymbol{\alpha}}) \in \mathcal{X}_{\text{Similar}}} |\boldsymbol{\omega}^{-1}(\boldsymbol{\alpha}) - \boldsymbol{\omega}^{-1}(\widehat{\boldsymbol{\alpha}})|\right) \sqrt{g(t_H)}}{\left(\min\limits_{(\boldsymbol{\beta},\widehat{\boldsymbol{\beta}}) \in \mathcal{X}_{\text{Dissimilar}}} \frac{1}{2}|\boldsymbol{\omega}^{-1}(\boldsymbol{\beta}) - \boldsymbol{\omega}^{-1}(\widehat{\boldsymbol{\beta}})|\right) \sqrt{g\left(\frac{1}{2}|t_{|A_1|+1} - t_1|\right)}}$$

$$\leq \lim_{\Delta \to +\infty} \frac{\left(\max\limits_{k \in \mathbb{N}_K} |A_k| - 1\right)}{\frac{1}{2}\Delta} \sqrt{\frac{g(t_H)}{g\left(\frac{1}{2}|t_{|A_1|+1} - t_1|\right)}}$$

$$\leq \lim_{\Delta \to +\infty} \frac{2\left(\max\limits_{k \in \mathbb{N}_K} |A_k| - 1\right)}{\Delta} \sqrt{\frac{g((K-1)\Delta + H)}{g\left(\frac{1}{2}\Delta + \frac{1}{2}|A_1|\right)}}$$

$$\leq \lim_{\Delta \to +\infty} \frac{2\left(\max\limits_{k \in \mathbb{N}_K} |A_k| - 1\right) (2(K-1))^{\varphi/2}}{\Delta} = 0, \tag{A.27}$$

where $\varphi$ is the order of the polynomial function $g(t) = \|\boldsymbol{\omega}'(t)\|_2^2$ and satisfies $0 \leq \varphi \leq 2(H + S) \leq 2(H + H(H-1))$. Further using the *non-negative* properties of $D^+$ and $D^-$, it holds that $\lim\limits_{\Delta \to +\infty} D^+/D^- = 0$. Therefore there exists the sufficiently large number $\Delta > 0$ such that

$$0 \leq \frac{D^+}{D^-} \leq \frac{1}{2}. \tag{A.28}$$

Let $m = \left\lceil \left(\frac{\Delta_{\text{margin}}}{D^+\text{Length}_{\boldsymbol{\omega}}(0,1)}\right)^2 \right\rceil$ and $\widetilde{\mathcal{M}}_{i::}(t) = \boldsymbol{\omega}(t)$ for $i = 1, 2, \cdots, m$, we obtain[4]

$$\text{Dist}_{\widetilde{\mathcal{M}}}(\boldsymbol{\beta}, \widehat{\boldsymbol{\beta}}) - \text{Dist}_{\widetilde{\mathcal{M}}}(\boldsymbol{\alpha}, \widehat{\boldsymbol{\alpha}})$$

$$= \sqrt{m\text{Length}_{\boldsymbol{\omega}}^2(0,1)\text{Length}_{\boldsymbol{\omega}}^2(\boldsymbol{\beta}, \widehat{\boldsymbol{\beta}})} - \sqrt{m\text{Length}_{\boldsymbol{\omega}}^2(0,1)\text{Length}_{\boldsymbol{\omega}}^2(\boldsymbol{\alpha}, \widehat{\boldsymbol{\alpha}})}$$

$$\geq \sqrt{m}\text{Length}_{\boldsymbol{\omega}}(0,1)(D^- - D^+)$$

$$\geq \sqrt{m}\text{Length}_{\boldsymbol{\omega}}(0,1)D^+$$

$$\geq \frac{\Delta_{\text{margin}}}{D^+\text{Length}_{\boldsymbol{\omega}}(0,1)}\text{Length}_{\boldsymbol{\omega}}(0,1)D^+$$

$$= \Delta_{\text{margin}}, \tag{A.29}$$

which completes the proof. $\square$

## B  Proof of Theorem 3 (Generalization Bound)

We firstly introduce the following lemmas for proving our Theorem 3 .

**Lemma 2** (McDiarmid's Inequality [3]). *Consider independent random variables $v_1, v_2, \cdots, v_n \in \mathcal{V}$ and a function $\phi : \mathcal{V}^n \to \mathbb{R}$. Suppose that for all $v_1, v_2, \cdots, v_n$ and $v_i' \in \mathcal{V}$ ($i = 1, 2, \cdots, n$), the function satisfies*

$$|\phi(v_1, \cdots, v_{i-1}, v_i, v_{i+1}, \cdots, v_n) - \phi(v_1, \cdots, v_{i-1}, v_i', v_{i+1}, \cdots, v_n)| \leq c_i, \quad \text{(B.1)}$$

*and then it holds that*

$$\mathcal{P}\{\phi(v_1, v_2, \cdots, v_n) - \mathbb{E}_{v_1, v_2, \cdots, v_n}(\phi(v_1, v_2, \cdots, v_n)) > \mu\} \leq e^{-\frac{2\mu^2}{\Sigma_{i=1}^n c_i^2}}. \quad \text{(B.2)}$$

**Lemma 3.** *Let $\mathcal{M}^* \in \mathbb{R}^{m \times d \times c}$ be the solution to the optimization objective*

$$\mathcal{M}^* \in \underset{\mathcal{M} \in \mathbb{R}^{m \times d \times c}}{\arg\min} \frac{1}{N} \sum_{j=1}^N \mathcal{L}(\text{Dist}_{\mathcal{M}}^2(\boldsymbol{x}_j, \widehat{\boldsymbol{x}}_j); y_j) + \lambda \|\mathcal{M}\|_F^2, \quad \text{(B.3)}$$

*then there exists a bounded tensor set $\mathcal{F}(\lambda)$ such that*

$$\mathcal{M}^* \in \mathcal{F}(\lambda) = \left\{ \mathcal{M} \,\Big|\, \mathcal{M}_{ijk} \in \left[ -\sqrt{\frac{C_0}{\lambda}}, \sqrt{\frac{C_0}{\lambda}} \right], i \in \mathbb{N}_m, j \in \mathbb{N}_d, \text{ and } k \in \mathbb{N}_c \right\}, \quad \text{(B.4)}$$

*where the constant $C_0 > 0$ is not dependent on $\lambda$.*

*Proof.* According to the optimality of $\mathcal{M}^*$, it follows that

$$\begin{aligned} &\frac{1}{N} \sum_{j=1}^N \mathcal{L}(\text{Dist}_{\mathcal{M}^*}^2(\boldsymbol{x}_j, \widehat{\boldsymbol{x}}_j); y_j) + \lambda \|\mathcal{M}^* - \boldsymbol{0}\|_F^2 \\ &\leq \frac{1}{N} \sum_{j=1}^N \mathcal{L}(\text{Dist}_{\boldsymbol{0}}^2(\boldsymbol{x}_j, \widehat{\boldsymbol{x}}_j); y_j) + \lambda \|\boldsymbol{0} - \boldsymbol{0}\|_F^2 \\ &\leq \frac{1}{N} \sum_{j=1}^N \mathcal{L}(\text{Dist}_{\boldsymbol{0}}^2(\boldsymbol{x}_j, \widehat{\boldsymbol{x}}_j); y_j). \end{aligned} \quad \text{(B.5)}$$

We denote that $\mathcal{L}_{\min} = \inf\limits_{\mathcal{M}^* \in \mathbb{R}^{m \times d \times c}, j=1,2,\cdots,N} \mathcal{L}(\text{Dist}_{\mathcal{M}^*}^2(\boldsymbol{x}_j, \widehat{\boldsymbol{x}}_j); y_j)$, and have that

$$\begin{aligned} &\lambda \|\mathcal{M}^* - \boldsymbol{0}\|_F^2 \\ &\leq \frac{1}{N} \sum_{j=1}^N \mathcal{L}(\text{Dist}_{\boldsymbol{0}}^2(\boldsymbol{x}_j, \widehat{\boldsymbol{x}}_j); y_j) - \frac{1}{N} \sum_{j=1}^N \mathcal{L}(\text{Dist}_{\mathcal{M}^*}^2(\boldsymbol{x}_j, \widehat{\boldsymbol{x}}_j); y_j) \\ &\leq \frac{1}{N} \sum_{j=1}^N \mathcal{L}(\text{Dist}_{\boldsymbol{0}}^2(\boldsymbol{x}_j, \widehat{\boldsymbol{x}}_j); y_j) - \frac{1}{N} \sum_{j=1}^N \mathcal{L}_{\min} \\ &= C_0, \end{aligned} \quad \text{(B.6)}$$

where $C_0 > 0$. Finally, we have

$$(\mathcal{M}_{ijk})^2 \leq \frac{C_0}{\lambda}, \quad \text{(B.7)}$$

which completes the proof. $\qquad\square$

The proof of Theorem 3 is given as follows.

**Theorem 3.** *Assume that $\mathcal{R}(\mathcal{M}) = \|\mathcal{M}\|_F^2 = \sum_{i,j,k}(\mathcal{M}_{ijk})^2$ and $\mathcal{M}^* \in \mathbb{R}^{m \times d \times c}$ is the solution to CDML. Then, we have that for any $0 < \delta < 1$ with probability $1 - \delta$*

$$\varepsilon(\mathcal{M}^*) - \bar{\varepsilon}_{\mathcal{X}}(\mathcal{M}^*) \leq X^* \sqrt{2\ln(1/\delta)/N} + B_\lambda R_N(\mathcal{L}), \quad \text{(B.8)}$$

*where $B_\lambda \to 0$ as $\lambda \to +\infty$. Here $R_N(\mathcal{L})$ is the Rademacher complexity[5] of the loss function $\mathcal{L}$ related to the space $\mathbb{R}^{m \times d \times c}$ for $N$ training pairs, and $X^* = \max_{k \in \mathbb{N}_N} \left| \mathcal{L}(\text{Dist}_{\mathcal{M}^*}^2(\boldsymbol{x}_k, \widehat{\boldsymbol{x}}_k); y_k) \right|$.*

*Proof.* Firstly, we denote that

$$\bar{\varepsilon}_{\mathcal{X},k}(\boldsymbol{\mathcal{M}}^*) = \bar{\varepsilon}_{\mathcal{X}}(\boldsymbol{\mathcal{M}}^*) - \frac{1}{N}(\mathcal{L}(\mathrm{Dist}^2_{\boldsymbol{\mathcal{M}}}(\boldsymbol{x}_k, \widehat{\boldsymbol{x}}_k); y_k) - \mathcal{L}(\mathrm{Dist}^2_{\boldsymbol{\mathcal{M}}}(\boldsymbol{x}, \widehat{\boldsymbol{x}}); y(\boldsymbol{x}, \widehat{\boldsymbol{x}}))), \qquad \text{(B.9)}$$

where $(\boldsymbol{x}, \widehat{\boldsymbol{x}}) \in \{(\boldsymbol{x}_j, \widehat{\boldsymbol{x}}_j) | j \in \mathbb{N}_N\}$ and $y(\boldsymbol{x}, \widehat{\boldsymbol{x}}) \in \{0, 1\}$ is the similarity label for $(\boldsymbol{x}, \widehat{\boldsymbol{x}})$. By Lemma 3, it follows that

$$
\begin{aligned}
&(\varepsilon(\boldsymbol{\mathcal{M}}^*) - \bar{\varepsilon}_{\mathcal{X}}(\boldsymbol{\mathcal{M}}^*)) - (\varepsilon(\boldsymbol{\mathcal{M}}^*) - \bar{\varepsilon}_{\mathcal{X},k}(\boldsymbol{\mathcal{M}}^*)) \\
&\leq |\bar{\varepsilon}(\boldsymbol{\mathcal{M}}^*) - \bar{\varepsilon}_{\mathcal{X},k}(\boldsymbol{\mathcal{M}}^*)| \\
&= \frac{1}{N} \left| \mathcal{L}(\mathrm{Dist}^2_{\boldsymbol{\mathcal{M}}^*}(\boldsymbol{x}_k, \widehat{\boldsymbol{x}}_k); y_k) - \mathcal{L}(\mathrm{Dist}^2_{\boldsymbol{\mathcal{M}}^*}(\boldsymbol{x}, \widehat{\boldsymbol{x}}); y(\boldsymbol{x}, \widehat{\boldsymbol{x}})) \right| \\
&\leq \frac{1}{N} \left( \left| \mathcal{L}(\mathrm{Dist}^2_{\boldsymbol{\mathcal{M}}^*}(\boldsymbol{x}_k, \widehat{\boldsymbol{x}}_k); y_k) \right| + \left| \mathcal{L}(\mathrm{Dist}^2_{\boldsymbol{\mathcal{M}}^*}(\boldsymbol{x}, \widehat{\boldsymbol{x}}); y(\boldsymbol{x}, \widehat{\boldsymbol{x}})) \right| \right) \\
&\leq \frac{2}{N} X^*,
\end{aligned}
\qquad \text{(B.10)}
$$

where $X^* = \max_{k \in \mathbb{N}_N} \left| \mathcal{L}(\mathrm{Dist}^2_{\boldsymbol{\mathcal{M}}^*}(\boldsymbol{x}_k, \widehat{\boldsymbol{x}}_k); y_k) \right|$. Then we apply Lemma 2 to the term $\varepsilon(\boldsymbol{\mathcal{M}}^*) - \bar{\varepsilon}_{\mathcal{X}}(\boldsymbol{\mathcal{M}}^*)$ and have that with probability $1 - \delta$ it holds that

$$\varepsilon(\boldsymbol{\mathcal{M}}^*) - \bar{\varepsilon}_{\mathcal{X}}(\boldsymbol{\mathcal{M}}^*) \leq \mathbb{E}_{\mathcal{X}} \left[ \varepsilon(\boldsymbol{\mathcal{M}}^*) - \bar{\varepsilon}_{\mathcal{X}}(\boldsymbol{\mathcal{M}}^*) \right] + X^* \sqrt{2\ln(1/\delta)/N}. \qquad \text{(B.11)}$$

Now we only need to estimate the first term of the right-hand side of the above inequality. Specifically, there holds

$$\mathbb{E}_{\mathcal{X}}[\varepsilon(\boldsymbol{\mathcal{M}}^*) - \bar{\varepsilon}_{\mathcal{X}}(\boldsymbol{\mathcal{M}}^*)] = \mathbb{E}_{\mathcal{X}}[\mathbb{E}_{\mathcal{Z}}(\bar{\varepsilon}_{\mathcal{Z}}(\boldsymbol{\mathcal{M}}^*)) - \bar{\varepsilon}_{\mathcal{X}}(\boldsymbol{\mathcal{M}}^*)] \leq \mathbb{E}_{\mathcal{X},\mathcal{Z}}[\bar{\varepsilon}_{\mathcal{Z}}(\boldsymbol{\mathcal{M}}^*) - \bar{\varepsilon}_{\mathcal{X}}(\boldsymbol{\mathcal{M}}^*)], \quad \text{(B.12)}$$

where $\mathcal{Z} = \{(\boldsymbol{z}_1, \widehat{\boldsymbol{z}}_1), (\boldsymbol{z}_2, \widehat{\boldsymbol{z}}_2), \cdots, (\boldsymbol{z}_N, \widehat{\boldsymbol{z}}_N) | (\boldsymbol{z}_j, \widehat{\boldsymbol{z}}_j) \sim \mathcal{D}, j \in \mathbb{N}_N\}$ are independent identically distributed (i.i.d.) samples which are independent of $\mathcal{X} = \{(\boldsymbol{x}_1, \widehat{\boldsymbol{x}}_1), (\boldsymbol{x}_2, \widehat{\boldsymbol{x}}_2), \cdots, (\boldsymbol{x}_N, \widehat{\boldsymbol{x}}_N) | (\boldsymbol{x}_j, \widehat{\boldsymbol{x}}_j) \sim \mathcal{D}, j \in \mathbb{N}_N\}$. By Lemma 3, we know that there exists the bounded tensor set $\mathcal{F}(\lambda)$ such that

$$\boldsymbol{\mathcal{M}}^* \in \mathcal{F}(\lambda) = \left\{ \boldsymbol{\mathcal{M}} | \mathcal{M}_{ijk} \in \left[ -\sqrt{\frac{C_0}{\lambda}}, \sqrt{\frac{C_0}{\lambda}} \right], i \in \mathbb{N}_m, j \in \mathbb{N}_d, \text{ and } k \in \mathbb{N}_c \right\}, \qquad \text{(B.13)}$$

where $C_0 > 0$ is a constant. Let the function

$$B_\lambda = 2\mathbb{E}_{\mathcal{X},\mathcal{Z}} \left[ \sup_{\boldsymbol{\mathcal{M}} \in \mathcal{F}(\lambda)} \bar{\varepsilon}_{\mathcal{Z}}(\boldsymbol{\mathcal{M}}) - \bar{\varepsilon}_{\mathcal{X}}(\boldsymbol{\mathcal{M}}) \right] / \mathbb{E}_{\mathcal{X},\mathcal{Z}} \left[ \sup_{\boldsymbol{\mathcal{M}} \in \mathbb{R}^{m \times d \times c}} \bar{\varepsilon}_{\mathcal{Z}}(\boldsymbol{\mathcal{M}}) - \bar{\varepsilon}_{\mathcal{X}}(\boldsymbol{\mathcal{M}}) \right]. \qquad \text{(B.14)}$$

By *Levi's Monotone Convergence Theorem* [1], we have

$$
\begin{aligned}
&\lim_{\lambda \to +\infty} \mathbb{E}_{\mathcal{X},\mathcal{Z}} \left[ \sup_{\boldsymbol{\mathcal{M}} \in \mathcal{F}(\lambda)} \bar{\varepsilon}_{\mathcal{Z}}(\boldsymbol{\mathcal{M}}) - \bar{\varepsilon}_{\mathcal{X}}(\boldsymbol{\mathcal{M}}) \right] \\
&= \mathbb{E}_{\mathcal{X},\mathcal{Z}} \left[ \lim_{\lambda \to +\infty} \sup_{\boldsymbol{\mathcal{M}} \in \mathcal{F}(\lambda)} \bar{\varepsilon}_{\mathcal{Z}}(\boldsymbol{\mathcal{M}}) - \sup_{\boldsymbol{\mathcal{M}} \in \mathcal{F}(\lambda)} \bar{\varepsilon}_{\mathcal{X}}(\boldsymbol{\mathcal{M}}) \right] \\
&= \mathbb{E}_{\mathcal{X},\mathcal{Z}} \left[ \bar{\varepsilon}_{\mathcal{Z}}(\boldsymbol{0}) - \bar{\varepsilon}_{\mathcal{X}}(\boldsymbol{0}) \right] \\
&= \mathbb{E}_{\mathcal{Z}} \left[ \bar{\varepsilon}_{\mathcal{Z}}(\boldsymbol{0}) \right] - \mathbb{E}_{\mathcal{X}} \left[ \bar{\varepsilon}_{\mathcal{X}}(\boldsymbol{0}) \right] \\
&= 0.
\end{aligned}
\qquad \text{(B.15)}
$$

Therefore, we obtain $\lim_{\lambda \to +\infty} B_\lambda = 0$. By standard symmetrization techniques for i.i.d. Rademacher variables $\boldsymbol{\sigma} = (\sigma_1, \sigma_2, \cdots, \sigma_N)^\top$, it follows that

$$
\mathbb{E}_{\mathcal{X},\mathcal{Z}} \left[ \bar{\varepsilon}_\mathcal{Z}(\mathcal{M}^*) - \bar{\varepsilon}_\mathcal{X}(\mathcal{M}^*) \right]
$$

$$
\leq \mathbb{E}_{\mathcal{X},\mathcal{Z}} \left[ \sup_{\mathcal{M} \in \mathcal{F}(\sqrt{C_0/\lambda}, 3\sqrt{C_0/\lambda})} \bar{\varepsilon}_\mathcal{Z}(\mathcal{M}) - \bar{\varepsilon}_\mathcal{X}(\mathcal{M}) \right]
$$

$$
= \frac{B_\lambda}{2} \mathbb{E}_{\mathcal{X},\mathcal{Z}} \left[ \sup_{\mathcal{M} \in \mathbb{R}^{m \times d \times c}} \bar{\varepsilon}_\mathcal{Z}(\mathcal{M}) - \bar{\varepsilon}_\mathcal{X}(\mathcal{M}) \right]
$$

$$
= \frac{B_\lambda}{2N} \mathbb{E}_{\mathcal{X},\mathcal{Z},\boldsymbol{\sigma}} \left[ \sup_{\mathcal{M}^* \in \mathbb{R}^{m \times d \times c}} \sum_{j=1}^N \sigma_i \left( \mathcal{L}(\mathrm{Dist}_\mathcal{M}^2(\boldsymbol{x}_j, \widehat{\boldsymbol{x}}_j)) - \mathcal{L}(\mathrm{Dist}_\mathcal{M}^2(\boldsymbol{z}_j, \widehat{\boldsymbol{z}}_j)) \right) \right]
$$

$$
= \frac{B_\lambda}{N} \mathbb{E}_{\mathcal{X},\boldsymbol{\sigma}} \left[ \sup_{\mathcal{M}^* \in \mathbb{R}^{m \times d \times c}} \sum_{j=1}^N \sigma_i \mathcal{L}(\mathrm{Dist}_\mathcal{M}^2(\boldsymbol{x}_j, \widehat{\boldsymbol{x}}_j)) \right]
$$

$$
= B_\lambda R_N(\mathcal{L}), \tag{B.16}
$$

where $\sigma_i \in \{-1, 1\}$ for $i = 1, 2, \cdots, n$, and $R_N(\mathcal{L})$ is the Rademacher complexity of $\mathcal{L}$. Finally, combining the above inequality with Eq. (B.11) and Eq. (B.12) completes the proof. □

## C Proof of Theorem 4 (Topological Property)

We firstly introduce the following Lemma 4 for proving our Theorem 4.

**Lemma 4.** *If the function* $\mathrm{Length}_{\boldsymbol{\theta}_i}(\boldsymbol{x}, \widehat{\boldsymbol{x}})$ *satisfies triangle property for* $i \in \mathbb{N}_m$, *then the curvilinear distance* $\mathrm{Dist}_\Theta(\boldsymbol{x}, \widehat{\boldsymbol{x}})$ *satisfies triangle property, where* $\Theta = (\boldsymbol{\theta}_1, \boldsymbol{\theta}_2, \cdots, \boldsymbol{\theta}_m)$.

*Proof.* For $i \in \mathbb{N}_m$ and $\boldsymbol{\alpha}, \boldsymbol{\beta}, \boldsymbol{\gamma} \in \mathbb{R}^d$, we assume that

$$
\mathrm{Length}_{\boldsymbol{\theta}_i}(\boldsymbol{\alpha}, \boldsymbol{\beta}) \leq \mathrm{Length}_{\boldsymbol{\theta}_i}(\boldsymbol{\alpha}, \boldsymbol{\gamma}) + \mathrm{Length}_{\boldsymbol{\theta}_i}(\boldsymbol{\gamma}, \boldsymbol{\beta}), \tag{C.1}
$$

and obtain that

$$
\mathrm{Length}_{\boldsymbol{\theta}_i}^2(\boldsymbol{\alpha}, \boldsymbol{\beta}) \leq \mathrm{Length}_{\boldsymbol{\theta}_i}^2(\boldsymbol{\alpha}, \boldsymbol{\gamma}) + \mathrm{Length}_{\boldsymbol{\theta}_i}^2(\boldsymbol{\gamma}, \boldsymbol{\beta}) + 2\mathrm{Length}_{\boldsymbol{\theta}_i}(\boldsymbol{\alpha}, \boldsymbol{\gamma})\mathrm{Length}_{\boldsymbol{\theta}_i}(\boldsymbol{\gamma}, \boldsymbol{\beta}), \tag{C.2}
$$

Accordingly, we have

$$
\mathrm{Dist}_\Theta^2(\boldsymbol{\alpha}, \boldsymbol{\beta})
$$

$$
= \sum_{i=1}^m s_{\boldsymbol{\theta}_i} \cdot \mathrm{Length}_{\boldsymbol{\theta}_i}^2(\boldsymbol{\alpha}, \boldsymbol{\gamma})
$$

$$
\leq \sum_{i=1}^m s_{\boldsymbol{\theta}_i} \cdot \mathrm{Length}_{\boldsymbol{\theta}_i}^2(\boldsymbol{\alpha}, \boldsymbol{\gamma}) + s_{\boldsymbol{\theta}_i} \mathrm{Length}_{\boldsymbol{\theta}_i}^2(\boldsymbol{\gamma}, \boldsymbol{\beta}) + 2s_{\boldsymbol{\theta}_i} \mathrm{Length}_{\boldsymbol{\theta}_i}^2(\boldsymbol{\alpha}, \boldsymbol{\gamma})\mathrm{Length}_{\boldsymbol{\theta}_i}^2(\boldsymbol{\gamma}, \boldsymbol{\beta})
$$

$$
= \mathrm{Dist}_\Theta^2(\boldsymbol{\alpha}, \boldsymbol{\gamma}) + \mathrm{Dist}_\Theta^2(\boldsymbol{\gamma}, \boldsymbol{\beta}) + 2\sum_{i=1}^m \left( \sqrt{s_{\boldsymbol{\theta}_i}} \mathrm{Length}_{\boldsymbol{\theta}_i}(\boldsymbol{\alpha}, \boldsymbol{\gamma}) \right) \left( \sqrt{s_{\boldsymbol{\theta}_i}} \mathrm{Length}_{\boldsymbol{\theta}_i}(\boldsymbol{\gamma}, \boldsymbol{\beta}) \right)
$$

$$
\leq \mathrm{Dist}_\Theta^2(\boldsymbol{\alpha}, \boldsymbol{\gamma}) + \mathrm{Dist}_\Theta^2(\boldsymbol{\gamma}, \boldsymbol{\beta}) + 2\sqrt{\sum_{i=1}^m s_{\boldsymbol{\theta}_i} \mathrm{Length}_{\boldsymbol{\theta}_i}^2(\boldsymbol{\alpha}, \boldsymbol{\gamma})} \sqrt{\sum_{i=1}^m s_{\boldsymbol{\theta}_i} \mathrm{Length}_{\boldsymbol{\theta}_i}^2(\boldsymbol{\gamma}, \boldsymbol{\beta})}
$$

$$
= \mathrm{Dist}_\Theta^2(\boldsymbol{\alpha}, \boldsymbol{\gamma}) + \mathrm{Dist}_\Theta^2(\boldsymbol{\gamma}, \boldsymbol{\beta}) + 2\mathrm{Dist}_\Theta(\boldsymbol{\alpha}, \boldsymbol{\gamma})\mathrm{Dist}_\Theta(\boldsymbol{\gamma}, \boldsymbol{\beta})
$$

$$
= \left( \mathrm{Dist}_\Theta(\boldsymbol{\alpha}, \boldsymbol{\gamma}) + \mathrm{Dist}_\Theta(\boldsymbol{\gamma}, \boldsymbol{\beta}) \right)^2, \tag{C.3}
$$

where the last "$\leq$" is based on the *Cauchy Inequality* [2]. Therefore, we obtain $\mathrm{Dist}_\Theta(\boldsymbol{\alpha}, \boldsymbol{\beta}) \leq \mathrm{Dist}_\Theta(\boldsymbol{\alpha}, \boldsymbol{\gamma}) + \mathrm{Dist}_\Theta(\boldsymbol{\gamma}, \boldsymbol{\beta})$, which completes the proof. □

**Theorem 4.** *For any learned curvilinear distance* $\mathrm{Dist}_\Theta(\boldsymbol{x}, \widehat{\boldsymbol{x}})$ *and its corresponding parameter* $\Theta$, *we denote* $\Theta'(\boldsymbol{\tau}) = (\boldsymbol{\theta}_1'(\tau_1), \boldsymbol{\theta}_2'(\tau_2), \cdots, \boldsymbol{\theta}_m'(\tau_m)) \in \mathbb{R}^{d \times m}$ *and have that*

*1).* $\mathrm{Dist}_\Theta(\boldsymbol{x}, \widehat{\boldsymbol{x}})$ *is a **pseudo-metric** for any* $\Theta \in \mathbb{F}_m$;

*2).* $\mathrm{Dist}_\Theta(\boldsymbol{x}, \widehat{\boldsymbol{x}})$ *is a **metric**, if* $\Theta'(\boldsymbol{\tau})$ *is full row rank for any* $\boldsymbol{\tau} = (\tau_1, \tau_2, \cdots, \tau_m)^\top \in \mathbb{R}^m$.

*Proof.*

**1).** According to the definition of curvilinear distance, it is obvious that $\mathrm{Dist}_{\Theta}(\boldsymbol{x}, \widehat{\boldsymbol{x}})$ satisfies the non-negativity. The symmetry property can also be validated, because switching $\boldsymbol{x}$ and $\widehat{\boldsymbol{x}}$ will not change the lower and upper limit of the integral, *i.e.*,

$$\mathrm{Dist}_{\Theta}(\boldsymbol{x}, \widehat{\boldsymbol{x}}) = \sqrt{\sum\nolimits_{i=1}^{m} s_{\boldsymbol{\theta}_i} \cdot \left( \int_{\min(\mathcal{T}_{\boldsymbol{\theta}_i}(\boldsymbol{x}), \mathcal{T}_{\boldsymbol{\theta}_i}(\widehat{\boldsymbol{x}}))}^{\max(\mathcal{T}_{\boldsymbol{\theta}_i}(\boldsymbol{x}), \mathcal{T}_{\boldsymbol{\theta}_i}(\widehat{\boldsymbol{x}}))} \|\boldsymbol{\theta}_i'(t)\|_2 dt \right)^2} = \mathrm{Dist}_{\Theta}(\widehat{\boldsymbol{x}}, \boldsymbol{x}). \qquad (\mathrm{C}.4)$$

By invoking Lemma 4, we only need to demonstrate the triangle property of $\mathrm{Length}_{\boldsymbol{\theta}_i}(\boldsymbol{x}, \widehat{\boldsymbol{x}})$. Actually, for any $\boldsymbol{\alpha}, \boldsymbol{\beta}, \boldsymbol{\gamma} \in \mathbb{R}^d$, there exist the following 3 cases and their corresponding results.

*(case-1).* $\mathcal{T}_{\boldsymbol{\theta}_i}(\boldsymbol{\gamma}) \leq \min\{\mathcal{T}_{\boldsymbol{\theta}_i}(\boldsymbol{\alpha}), \mathcal{T}_{\boldsymbol{\theta}_i}(\boldsymbol{\beta})\}$:

$$\begin{aligned}
&\mathrm{Length}_{\boldsymbol{\theta}_i}(\boldsymbol{\alpha}, \boldsymbol{\gamma}) + \mathrm{Length}_{\boldsymbol{\theta}_i}(\boldsymbol{\gamma}, \boldsymbol{\beta}) \\
&= \int_{\min(\mathcal{T}_{\boldsymbol{\theta}_i}(\boldsymbol{\alpha}), \mathcal{T}_{\boldsymbol{\theta}_i}(\boldsymbol{\gamma}))}^{\max(\mathcal{T}_{\boldsymbol{\theta}_i}(\boldsymbol{\alpha}), \mathcal{T}_{\boldsymbol{\theta}_i}(\boldsymbol{\gamma}))} \|\boldsymbol{\theta}_i'(t)\|_2 \, dt + \int_{\min(\mathcal{T}_{\boldsymbol{\theta}_i}(\boldsymbol{\gamma}), \mathcal{T}_{\boldsymbol{\theta}_i}(\boldsymbol{\beta}))}^{\max(\mathcal{T}_{\boldsymbol{\theta}_i}(\boldsymbol{\gamma}), \mathcal{T}_{\boldsymbol{\theta}_i}(\boldsymbol{\beta}))} \|\boldsymbol{\theta}_i'(t)\|_2 \, dt \\
&\geq \int_{\mathcal{T}_{\boldsymbol{\theta}_i}(\boldsymbol{\gamma})}^{\mathcal{T}_{\boldsymbol{\theta}_i}(\boldsymbol{\alpha})} \|\boldsymbol{\theta}_i'(t)\|_2 \, dt + \int_{\mathcal{T}_{\boldsymbol{\theta}_i}(\boldsymbol{\gamma})}^{\mathcal{T}_{\boldsymbol{\theta}_i}(\boldsymbol{\beta})} \|\boldsymbol{\theta}_i'(t)\|_2 \, dt \\
&\geq \int_{\min(\mathcal{T}_{\boldsymbol{\theta}_i}(\boldsymbol{\alpha}), \mathcal{T}_{\boldsymbol{\theta}_i}(\boldsymbol{\beta}))}^{\max(\mathcal{T}_{\boldsymbol{\theta}_i}(\boldsymbol{\alpha}), \mathcal{T}_{\boldsymbol{\theta}_i}(\boldsymbol{\beta}))} \|\boldsymbol{\theta}_i'(t)\|_2 \, dt \\
&= \mathrm{Length}_{\boldsymbol{\theta}_i}(\boldsymbol{\alpha}, \boldsymbol{\beta}). \qquad (\mathrm{C}.5)
\end{aligned}$$

*(case-2).* $\min\{\mathcal{T}_{\boldsymbol{\theta}_i}(\boldsymbol{\alpha}), \mathcal{T}_{\boldsymbol{\theta}_i}(\boldsymbol{\beta})\} < \mathcal{T}_{\boldsymbol{\theta}_i}(\boldsymbol{\gamma}) < \max\{\mathcal{T}_{\boldsymbol{\theta}_i}(\boldsymbol{\alpha}), \mathcal{T}_{\boldsymbol{\theta}_i}(\boldsymbol{\beta})\}$:

$$\begin{aligned}
&\mathrm{Length}_{\boldsymbol{\theta}_i}(\boldsymbol{\alpha}, \boldsymbol{\gamma}) + \mathrm{Length}_{\boldsymbol{\theta}_i}(\boldsymbol{\gamma}, \boldsymbol{\beta}) \\
&= \int_{\min(\mathcal{T}_{\boldsymbol{\theta}_i}(\boldsymbol{\alpha}), \mathcal{T}_{\boldsymbol{\theta}_i}(\boldsymbol{\gamma}))}^{\max(\mathcal{T}_{\boldsymbol{\theta}_i}(\boldsymbol{\alpha}), \mathcal{T}_{\boldsymbol{\theta}_i}(\boldsymbol{\gamma}))} \|\boldsymbol{\theta}_i'(t)\|_2 \, dt + \int_{\min(\mathcal{T}_{\boldsymbol{\theta}_i}(\boldsymbol{\gamma}), \mathcal{T}_{\boldsymbol{\theta}_i}(\boldsymbol{\beta}))}^{\max(\mathcal{T}_{\boldsymbol{\theta}_i}(\boldsymbol{\gamma}), \mathcal{T}_{\boldsymbol{\theta}_i}(\boldsymbol{\beta}))} \|\boldsymbol{\theta}_i'(t)\|_2 \, dt \\
&= \int_{\min(\mathcal{T}_{\boldsymbol{\theta}_i}(\boldsymbol{\alpha}), \mathcal{T}_{\boldsymbol{\theta}_i}(\boldsymbol{\beta}))}^{\mathcal{T}_{\boldsymbol{\theta}_i}(\boldsymbol{\gamma})} \|\boldsymbol{\theta}_i'(t)\|_2 \, dt + \int_{\mathcal{T}_{\boldsymbol{\theta}_i}(\boldsymbol{\gamma})}^{\max(\mathcal{T}_{\boldsymbol{\theta}_i}(\boldsymbol{\alpha}), \mathcal{T}_{\boldsymbol{\theta}_i}(\boldsymbol{\beta}))} \|\boldsymbol{\theta}_i'(t)\|_2 \, dt \\
&= \mathrm{Length}_{\boldsymbol{\theta}_i}(\boldsymbol{\alpha}, \boldsymbol{\beta}). \qquad (\mathrm{C}.6)
\end{aligned}$$

*(case-3).* $\mathcal{T}_{\boldsymbol{\theta}_i}(\boldsymbol{\gamma}) \geq \max\{\mathcal{T}_{\boldsymbol{\theta}_i}(\boldsymbol{\alpha}), \mathcal{T}_{\boldsymbol{\theta}_i}(\boldsymbol{\beta})\}$:

$$\begin{aligned}
&\mathrm{Length}_{\boldsymbol{\theta}_i}(\boldsymbol{\alpha}, \boldsymbol{\gamma}) + \mathrm{Length}_{\boldsymbol{\theta}_i}(\boldsymbol{\gamma}, \boldsymbol{\beta}) \\
&= \int_{\min(\mathcal{T}_{\boldsymbol{\theta}_i}(\boldsymbol{\alpha}), \mathcal{T}_{\boldsymbol{\theta}_i}(\boldsymbol{\gamma}))}^{\max(\mathcal{T}_{\boldsymbol{\theta}_i}(\boldsymbol{\alpha}), \mathcal{T}_{\boldsymbol{\theta}_i}(\boldsymbol{\gamma}))} \|\boldsymbol{\theta}_i'(t)\|_2 \, dt + \int_{\min(\mathcal{T}_{\boldsymbol{\theta}_i}(\boldsymbol{\gamma}), \mathcal{T}_{\boldsymbol{\theta}_i}(\boldsymbol{\beta}))}^{\max(\mathcal{T}_{\boldsymbol{\theta}_i}(\boldsymbol{\gamma}), \mathcal{T}_{\boldsymbol{\theta}_i}(\boldsymbol{\beta}))} \|\boldsymbol{\theta}_i'(t)\|_2 \, dt \\
&\geq \int_{\mathcal{T}_{\Theta_i}(\boldsymbol{\alpha})}^{\mathcal{T}_{\Theta_i}(\boldsymbol{\gamma})} \|\boldsymbol{\Theta}_i'(t)\|_2 \, dt + \int_{\mathcal{T}_{\boldsymbol{\theta}_i}(\boldsymbol{\beta})}^{\mathcal{T}_{\boldsymbol{\theta}_i}(\boldsymbol{\gamma})} \|\boldsymbol{\theta}_i'(t)\|_2 \, dt \\
&\geq \int_{\min(\mathcal{T}_{\boldsymbol{\theta}_i}(\boldsymbol{\alpha}), \mathcal{T}_{\boldsymbol{\theta}_i}(\boldsymbol{\beta}))}^{\max(\mathcal{T}_{\boldsymbol{\theta}_i}(\boldsymbol{\alpha}), \mathcal{T}_{\boldsymbol{\theta}_i}(\boldsymbol{\beta}))} \|\boldsymbol{\theta}_i'(t)\|_2 \, dt \\
&= \mathrm{Length}_{\boldsymbol{\theta}_i}(\boldsymbol{\alpha}, \boldsymbol{\beta}). \qquad (\mathrm{C}.7)
\end{aligned}$$

From the results of above 3 cases, we know that the triangle property is satisfied for any $\boldsymbol{\Theta} \in \mathbb{F}_m$ and thus the proof is completed.

**2).** It is obvious that for any $\boldsymbol{x}, \widehat{\boldsymbol{x}} \in \mathbb{R}^d$,

$$\boldsymbol{x} = \widehat{\boldsymbol{x}} \implies \mathrm{Dist}_{\Theta}(\boldsymbol{x}, \widehat{\boldsymbol{x}}) = 0. \qquad (\mathrm{C}.8)$$

We just need to prove that $\boldsymbol{x} = \widehat{\boldsymbol{x}}$ for $\mathrm{Dist}_{\Theta}(\boldsymbol{x}, \widehat{\boldsymbol{x}}) = 0$. Assume that $\mathrm{Rank}(\boldsymbol{\Theta}'(\boldsymbol{\tau})) = d$, we obtain $\boldsymbol{\theta}_i'(\frac{1}{2}) \neq 0$. The scale value $\mathrm{Length}_{\boldsymbol{\theta}_i}(0, 1)$ satisfies

$$\mathrm{Length}_{\boldsymbol{\theta}_i}(0, 1) = \int_0^1 \|\boldsymbol{\theta}_i'(t)\|_2 \, dt \geq \int_{\frac{1}{2}-\epsilon}^{\frac{1}{2}+\epsilon} \|\boldsymbol{\theta}_i'(t)\|_2 dt > 0, \qquad (\mathrm{C}.9)$$

where $0 < \epsilon < \frac{1}{2}$ is a sufficiently small number such that $\boldsymbol{\theta}_i'(t) \neq 0$ for $t \in (\frac{1}{2} - \epsilon, \frac{1}{2} + \epsilon)$. We thus have

$$\mathrm{Dist}_{\Theta}(\boldsymbol{x}, \widehat{\boldsymbol{x}}) = 0 \implies \mathrm{Length}_{\boldsymbol{\theta}_i}(0, 1)\mathrm{Length}_{\boldsymbol{\theta}_i}(\boldsymbol{x}, \widehat{\boldsymbol{x}}) = 0 \implies \mathrm{Length}_{\boldsymbol{\theta}_i}(\boldsymbol{x}, \widehat{\boldsymbol{x}}) = 0, \qquad (\mathrm{C}.10)$$

where $i \in \mathbb{N}_m$. Therefore, we have $\mathcal{T}_{\boldsymbol{\theta}_i}(\boldsymbol{x}) = \mathcal{T}_{\boldsymbol{\theta}_i}(\widehat{\boldsymbol{x}})$. According to the definition of the calibration point, it follows that

$$\mathcal{T}_{\boldsymbol{\theta}_i}(\boldsymbol{x}) \in \arg\min_{t \in \mathbb{R}} \|\boldsymbol{\theta}_i(t) - \boldsymbol{x}\|_2^2, \tag{C.11}$$

and

$$\mathcal{T}_{\boldsymbol{\theta}_i}(\widehat{\boldsymbol{x}}) \in \arg\min_{t \in \mathbb{R}} \|\boldsymbol{\theta}_i(t) - \widehat{\boldsymbol{x}}\|_2^2. \tag{C.12}$$

Namely, it holds that

$$\mathcal{T}_{\boldsymbol{\theta}_i}(\boldsymbol{x}) \in \{t|\, (\boldsymbol{\theta}_i(t) - \boldsymbol{x})^\top \boldsymbol{\theta}_i'(t) = 0\}, \tag{C.13}$$

and

$$\mathcal{T}_{\boldsymbol{\theta}_i}(\widehat{\boldsymbol{x}}) \in \{t|\, (\boldsymbol{\theta}_i(t) - \widehat{\boldsymbol{x}})^\top \boldsymbol{\theta}_i'(t) = 0\}. \tag{C.14}$$

Since $\mathcal{T}_{\boldsymbol{\theta}_i}(\boldsymbol{x}) = \mathcal{T}_{\boldsymbol{\theta}_i}(\widehat{\boldsymbol{x}}) = \tau_i$, we have the following equation group

$$\begin{cases} (\boldsymbol{\theta}_i(\tau_i) - \boldsymbol{x})^\top \boldsymbol{\theta}_i'(\tau_i) = 0, \\ (\boldsymbol{\theta}_i(\tau_i) - \widehat{\boldsymbol{x}})^\top \boldsymbol{\theta}_i'(\tau_i) = 0. \end{cases} \tag{C.15}$$

The equation difference of Eq. (C.15) gives that

$$\boldsymbol{\theta}_i'(\tau_i)^\top (\boldsymbol{x} - \widehat{\boldsymbol{x}}) = 0. \tag{C.16}$$

For $i \in \mathbb{N}_m$, we thus have the linear equation group *w.r.t.* $\boldsymbol{x} - \widehat{\boldsymbol{x}}$

$$\left(\boldsymbol{\theta}_1'(\tau_1), \boldsymbol{\theta}_2'(\tau_2), \cdots, \boldsymbol{\theta}_m'(\tau_m)\right)^\top (\boldsymbol{x} - \widehat{\boldsymbol{x}}) = \boldsymbol{0}. \tag{C.17}$$

As $\mathrm{Rank}(\boldsymbol{\Theta}'(\boldsymbol{\tau})) = \mathrm{Rank}\left(\boldsymbol{\theta}_1'(\tau_1), \boldsymbol{\theta}_2'(\tau_2), \cdots, \boldsymbol{\theta}_m'(\tau_m)\right) = d$, we know that the above Eq. (C.17) has the unique solution $\boldsymbol{x} - \widehat{\boldsymbol{x}} = \boldsymbol{0}$. Therefore, $\boldsymbol{x} = \widehat{\boldsymbol{x}}$ holds for $\mathrm{Dist}_{\boldsymbol{\Theta}}(\boldsymbol{x}, \widehat{\boldsymbol{x}}) = 0$, which completes the proof. □

## Footnotes

[1] Here $\kappa(j)$ denotes the maximal integer satisfying $\sum_{i=1}^{\kappa(j)} |A_j| < j$.

[2]Due to $(f_{\boldsymbol{\mu}_1}(\widetilde{t}_j), f_{\boldsymbol{\mu}_2}(\widetilde{t}_j), \cdots, f_{\boldsymbol{\mu}_{d-1}}(\widetilde{t}_j)) = (b_{l(j)1}, b_{l(j)2}, \cdots, b_{l(j)(d-1)})$ and $\widetilde{f}_{\boldsymbol{\mu}_d}(\widetilde{t}_j) \neq b_{l(j)d}$.

[3]There is no function $l(j)$ satisfying $(f_{\boldsymbol{\mu}_1}(\widetilde{t}_j), f_{\boldsymbol{\mu}_2}(\widetilde{t}_j), \cdots, f_{\boldsymbol{\mu}_{d-1}}(\widetilde{t}_j)) = (b_{l(j)1}, b_{l(j)2}, \cdots, b_{l(j)(d-1)})$.

[4] Here the operator $\lceil a \rceil$ denotes the smallest integer that is not smaller than $a$.

[5]The Rademacher complexity of the hypothesis $f$ over the space $\mathcal{F}$ is defined as $R_N(f) = \mathbb{E}_{\mathcal{X}, \boldsymbol{\sigma}}[\sup_{\mathcal{M} \in \mathcal{F}} \frac{1}{N} \sum_{j=1}^N \sigma_j f(\mathcal{M})]$, where $\mathcal{X} = \{(\boldsymbol{x}_j, \widehat{\boldsymbol{x}}_j) \sim \mathcal{D} | j \in \mathbb{N}_N\}$, and $P\{\sigma_j = -1\} = P\{\sigma_j = 1\} = 0.5$ for $j \in \mathbb{N}_N$.