[Reviews · NeurIPS 2019]

Reviewer 1



The paper is very well written. The motivation is clear and the approach which involves finding curved axis for projecting the data is very intuitive. The authors provide theoretical analysis that significantly improve the quality of the paper. The experimental evaluation is also convincing. Overall, I believe this is a significant contribution and it should appear in NeurIPS. My main questions/comments are as follows: 1) Shouldn't there be a notion of orthonormality for the curved axis (thetas)? How do you enforce the axis from collapsing to directions with higher variation? 2) The discussion in page on derivation of the (smoothed) gradient is a bit unclear and hard to follow. Please try to simplify that. 3) Can you please comment on the runtime and the complexity of the algorithm? How does it scales to significantly large datasets? It would have been nice to have access to the code for having a better assessment of the method.

Reviewer 2



The paper is well written in general. My main concern is that, although the formulation of the proposed metric is new to my knowledge, the choice of baselines is not appropriate. The proposed model is a NONLINEAR generalization of the linear Mahalanobis. However, most of the baseline methods (except DDML and PML) are standard LINEAR models. This comparison is unfair since it is known that nonlinear approaches (e.g. nonlinear kernels or neural networks) outperform linear models in metric learning. DDML was published in CVPR 2014 and PML was published in CVPR 2015. They then do not correspond to state-of-the-art deep metric learning approaches. The authors could use more recent baselines such as those reported in [A] (angular loss etc...). The authors also do not explain how they trained the nonlinear baselines. Did the authors use the neural network architectures used in the original papers? How were the hyperparameters determined etc... [A] Zhai and Wu, Making Classification Competitive for Deep Metric Learning, 2018

Reviewer 3



Originality: The method is new and provides a direct generalization of the Linear Distance Metric learning. Quality: Theorems are clearly interesting to validate the methodology. Fitting capacity result (Theorem 2) ensures that there exists a curvilinear metric that can well separate the data. The Generalization bound ensures empirical loss converges to the expected loss. However, it is unclear whether this ensures that the algorithm converges to the/a Distance introduced by Theorem 2 (the distance well separating the data). It does not seem to ensure that the obtained Distance perfom well either. Also, a discussion on the purpose of tensor $A$ in Theorem 3 could be beneficial. Finally, the topological property certifies that the object learned is a (pseudo)metric; which is a useful property. Clarity: The paper is well written and well organised and pleasant to read. In particular, the "Geometric interpretation" paragraph is a good step to get the insights of the paper. However, it should be stated clearly that the authors focus on "supervised metric learning", aimed at learning metric for classification; the first 3 pages are quite misleading in that matter. Significance: The comparison with the state of the art results convince me that it is a method likely to be used in practice. Although the theoretical results ensures polynomial approximation of the metric could behave well enough for classification, it is not clear whether the algorithm converges to this solution after reading Theorem 3.

[Author Response · NeurIPS 2019]

We thank the three reviewers for their constructive comments. The following are our responses to reviewers' comments.

**—To Reviewer #1—**

**Re. the notion of orthonormality:** When we designed our method, we followed the original forms of existing metric
learning models and thus did not use the additional orthonormality constraints. Actually, metric learning aims to find
embedding directions (see Fig. 1 in our paper) so that the resulting metric can faithfully preserve the intrinsic distances
of data pairs. The directions with necessarily high variations for the subsequent classification are usually favored by
many dimension reduction techniques such as PCA.

**Re. simplifying derivations and proofs:** As the reviewer suggested, in the final paper, we will try our best to simplify
the derivations of gradients, and carefully expand the proofs to make them easier to understand.

**Re. the complexity, runtime, and code release:** The matrix multiplication complexities of Eq. (14) and Eq. (16) are
$O(c^2hmd)$ and $O(Lchmd)$, respectively. Here $h$ and $d$ are the batch-size and data dimension correspondingly, and the
constants $c$ and $L$ are independent of the size of datasets. Since the measurer line number $m$ is always set to be smaller
than $d$, the total complexity of our algorithm is $O(hd^2)$, which is the same as most of the baseline methods. The results
of CPU hours (Core Duo 2.93GHz desktop with 16G RAM) on MVS dataset ($10^5$ training pairs and $10^4$ test pairs) are
presented in Table I, which show that our method requires comparable runtime with existing methods. We will release
the code if this paper is accepted.

Table I: Runtime comparisons.

| Methods | Training | Test |
|---|---|---|
| LMNN | 2.35h | 0.12h |
| ITML | 1.96h | 0.12h |
| DDML | 3.65h | 0.18h |
| PML | 1.92h | 0.15h |
| ODML | 2.12h | 0.12h |
| BDML | 2.69h | 0.12h |
| **CDML** | 2.59h | 0.16h |

Table II: Classification error rates (%) (lower is better) and verification AUC values (higher is better) of compared methods.

| Methods | Classification Tasks | | | | | | Verification Tasks | | |
|---|---|---|---|---|---|---|---|---|---|
| | Letter | Autompg | Australia | Glass | Balance | Segment | Pub. | LFW | MVS |
| Npairs | 4.32±0.51• | 25.94±3.64• | 16.32±0.12• | 27.29±0.24• | 9.09±0.12• | 7.13±2.32• | 90.1 | 87.9 | 73.7 |
| Angular | 3.55±0.61• | 23.24±0.73• | 15.32±2.56 | 28.12±0.23• | 8.19±0.64• | 6.19±3.64• | 91.6 | 88.8 | 72.4 |
| DAML | 3.21±0.66• | 22.23±0.61• | 17.09±1.14• | 24.12±3.54 | 9.03±0.64• | 4.03±0.89• | 91.5 | 88.1 | 76.6 |
| Hard-Aware | 3.11±0.23• | 22.92±1.14• | 16.32±1.14• | **22.09±5.64** | 8.14±1.02 | 5.24±2.65• | 91.9 | 88.5 | 73.2 |
| **CDML** | **2.09±0.64** | **15.32±6.11** | **12.22±2.54** | 22.12±4.64 | **5.01±2.64** | **1.23±0.32** | **92.4** | **89.1** | **77.1** |

**—To Reviewer #2—**

**Re. the interpretation should be regarded as a good motivation:** Thanks for your suggestions. We will modify our
claim on interpretation (*i.e.,* Line 101) from the viewpoint of motivation. "which might be more intuitive than the
previous interpretations." ⟶ "which offers a clear way to handle the nonlinear data with geometric structures."

**Re. more recent baselines and DDML training details:** As the reviewer suggested, we add new experiments for
comparing the baseline methods "Npairs Loss" and "Angular Loss" reported in "Making Classification Competitive
for Deep Metric Learning"(*arXiv 2018*, recommended by the reviewer). We also add two latest baselines "Deep
Asymmetric Metric Learning(DAML) via Rich Relationship Mining"(*CVPR 2019*) and "Hardness-Aware Deep Metric
Learning"(*CVPR 2019*) for further comparisons. The six classification datasets and three verification datasets in our
paper are used here. Table II lists the error rates on classification tasks ("•" denotes a significantly better result at the
significance level 0.05) and AUC values on verification tasks for various methods. Obviously, our CDML outperforms
the recent baselines in most cases. We believe that the above new results further improve the fairness and sufficiency of
our experiments, and we will duly add them in our final paper. For the training of DDML details, the regularization
parameter $\lambda$ was tuned via searching the grid $\{10^{-2}, 10^{-1}, 1, 10, 10^2\}$ by observing the model performance on validation
set. Other configurations such as network architectures, weight initializations, and SGD-related parameters were set as
recommended by the authors of "Discriminative Deep Metric Learning for Face Verification in the Wild"(*CVPR 2014*).

**—To Reviewer #3—**

**Re. the solution and Theorem 2/3:** The theoretical analyses on generalization bound usually focus on the ideal case
when the globally optimal solution is obtained, although the models are nonconvex such as "Learning Latent Space
Models with Angular Constraints"(*ICML 2017*) and "Fast Generalization Error Bound of Deep Learning from a Kernel
Perspective"(*ICML 2018*). We thus follow such common practice and also discuss the ideal case in our theoretical
analysis. The globally optimal solution might not be acquired by our method practically due to the non-convexity of
objective function, and this practical phenomenon is also observed in above prior works.

**Re. discussing tensor $\mathcal{A}$ and function $B(\lambda)$:** The tensor $\mathcal{A}$ is predefined to smooth and stabilize the learning of
polynomial coefficients. We can treat it as a constant which restricts the high variations of the learning parameter
$\mathcal{M}$ within a small hypothesis space. In our experiments, $\mathcal{A}$ is simply fixed to $\mathbf{0}$, *i.e.,* using the original Frobenius-
norm regularizer. For the function $B(\lambda)$, its expression has been shown in Eq. (B.14) in supplemental materials as
$B(\lambda) = 2\mathbb{E}_{\mathcal{X},\mathcal{Z}}(\sup_{\mathcal{M}\in\mathcal{F}(\lambda)}\overline{\varepsilon}_{\mathcal{Z}}(\mathcal{M}) - \overline{\varepsilon}_{\mathcal{X}}(\mathcal{M}))/\mathbb{E}_{\mathcal{X},\mathcal{Z}}(\sup_{\mathcal{M}\in\mathbb{R}^{m\times d\times c}}\overline{\varepsilon}_{\mathcal{Z}}(\mathcal{M}) - \overline{\varepsilon}_{\mathcal{X}}(\mathcal{M}))$. This expression reveals that
when the regularization parameter $\lambda$ increases, the hypothesis space $\mathcal{F}(\lambda)$ shrinks, so the numerator in the above
expression decreases (the denominator does not change as it is irrelevant to $\lambda$), which further leads to a smaller $B(\lambda)$
and a tighter upper bound. We will add the above discussions in the final paper.

**Re. declaration for "supervised metric learning":** Thanks. We will carefully declare that our paper focuses on
supervised metric learning in the Introduction section.

[Meta-Review · NeurIPS 2019]

Thanks for your submission to NeurIPS. This paper had somewhat mixed reviews, with two positive reviews and one negative review. After the rebuttal, I also took a look at the paper and initiated a discussion with the reviewers. The main concerns of the negative reviewer were in the comparison to existing non-linear metric learning approaches; I also had a similar concern when reading the paper. I think the rebuttal responded well to this (also, it's unrealistic that one could compare with many of the existing non-linear approaches, given how many exist); please include the new results in the final version. My other concern was that the writing is often not super clear---for example, terms like "measurer lines" are not ever precisely defined, and may be unfamiliar to readers who work on metric learning. Whatever can be done for the final version to clarify the text would make for a more effective presentation. Generally speaking, please keep in mind all the reviewer comments when preparing a final version.